# Association of Preoperative Prognostic Nutritional Index and Postoperative Acute Kidney Injury in Patients with Colorectal Cancer Surgery

**DOI:** 10.3390/nu13051604

**Published:** 2021-05-11

**Authors:** Ji-Hoon Sim, Ji-Yeon Bang, Sung-Hoon Kim, Sa-Jin Kang, Jun-Gol Song

**Affiliations:** Department of Anesthesiology and Pain Medicine, Asan Medical Center, University of Ulsan College of Medicine, Seoul 05505, Korea; atlassjh@hanmail.net (J.-H.S.); jyounbang@gmail.com (J.-Y.B.); shkimans@gmail.com (S.-H.K.); sajinkg@naver.com (S.-J.K.)

**Keywords:** acute kidney injury, colorectal cancer, prognostic nutritional index, survival

## Abstract

The prognostic nutritional index (PNI) has been reported to be associated with postoperative complications and prognosis in cancer surgery. However, few studies have evaluated the association between preoperative PNI and postoperative acute kidney injury (AKI) in colorectal cancer patients. This study evaluated association of preoperative PNI and postoperative AKI in patients who underwent colorectal cancer surgery. This study retrospectively analyzed 3543 patients who underwent colorectal cancer surgery between June 2008 and February 2012. The patients were classified into four groups by the quartile of PNI: Q1 (≤43.79), Q2 (43.79–47.79), Q3 (47.79–51.62), and Q4 (≥51.62). Multivariate regression analysis was performed to assess the risk factors for AKI and 1-year mortality. AKI was defined according to Kidney Disease Improving Global Outcomes classification (KDIGO) criteria. Additionally, we assessed surgical outcomes such as hospital stay, ICU admission, and postoperative complications. The incidence of postoperative AKI tended to increase in the Q1 group (13.4%, 9.2%, 9.4%, 8.8%). In the multivariate analysis, high preoperative PNI was significantly associated with low risk of postoperative AKI (adjusted odds ratio [OR]: 0.96, 95% confidence interval [CI]: 0.93–0.99, *p* = 0.003) and low 1-year mortality (OR: 0.92, 95% CI: 0.86–0.98, *p* = 0.011). Male sex, body mass index, diabetes mellitus, and hypertension were risk factors for AKI. The Q1 (≤43.79) group had poor surgical outcomes, such as postoperative AKI (OR: 1.52, 95% CI: 1.18–1.95, *p* = 0.001), higher rates of ICU admission (OR: 3.13, 95% CI: 1.82–5.39, *p* < 0.001) and higher overall mortality (OR: 3.81, 95% CI: 1.86–7.79, *p* < 0.001). In conclusion, low preoperative PNI levels, especially in the Q1 (≤43.79), were significantly associated with postoperative AKI and surgical outcomes, such as hospital stay, postoperative ICU admission, and mortality.

## 1. Introduction

Acute kidney injury (AKI) is one of common postoperative complications in colorectal cancer patients. The incidence of AKI in patients with colorectal cancer surgery has been reported to be 3.8–20.8% [1,2]. AKI increases the risk of developing chronic kidney disease (CKD) and end stage renal disease (ESRD) [3] and increases morbidity, mortality, and hospital cost [4,5]. Therefore, there have been many studies to determine the risk factors of postoperative AKI in colorectal cancer patients [1,6,7].

The prognostic nutritional index (PNI), which is calculated from the serum albumin concentration and the total lymphocyte count in the peripheral blood [8], reflects nutritional and inflammatory status [9]. Malnutrition has a significant impact on patient prognosis by reducing immune function, and lymphocyte count is an indicator of cell-mediated immunity and plays an important role in cancer prevention [8,9]. Therefore, more attention has been paid to the impact of nutritional and immune status on the prognosis of various cancers. PNI has been reported as a simple and objective indicator of patient survival and complications in acute and chronic diseases, such as AKI, heart failure, and various cancers [10,11,12,13,14]. Moreover, PNI was reported to be associated with postoperative complications and surgical prognosis in patients with colorectal cancer surgery [15,16,17,18].

However, to our knowledge, no studies have evaluated the association between preoperative PNI and postoperative AKI development in colorectal cancer patients. Therefore, this study was performed to evaluate the association preoperative PNI and postoperative AKI in patients who underwent colorectal cancer surgery. We also assessed hospital days, postoperative intensive care unit (ICU) admission and complications, and 1-year and overall mortality.

## 2. Materials and Methods

### 2.1. Study Design and Patient Population

The institutional review board (IRB) of the Asan Medical Center (protocol number: 2020-1806) approved this retrospective study, and the requirement for written informed consent was waived by the IRB. Based on the 10th revision of the International Classification of Diseases (ICD-10), we reviewed all patients diagnosed with colorectal cancer and undergoing surgery between June 2008 and February 2012. Adult patients over 18 years of age were included in the study.

The exclusion criteria were as follows: Patients with severe cardiopulmonary diseases; patients diagnosed with chronic kidney disease; patients who had already undergone kidney replacement therapy; patients requiring interventions pertaining to urethra, ureter, or kidneys during operation; patients who underwent emergency surgery; and patients with incomplete data or laboratory values.

### 2.2. General Anesthesia and Surgical Technique

Regardless of the surgical approach, all patients underwent general anesthesia (open vs. laparoscopy). All surgeries were performed by a qualified and experienced surgical team. Conventional and laparoscopic surgeries were performed using standard protocols [19]. For laparoscopic surgery, pneumoperitoneum with carbon dioxide at 10–15 mmHg was established. Crystalloid and colloid solutions were administered during surgery. The total volume of infused synthetic colloids did not exceed 20 mL kg^−1^. During surgery, packed red blood cells (RBC) were transfused if plasma hemoglobin level was less than 8 g dL^−1^, and inotropes or vasopressors were administered if the mean blood pressure was less than 65 mmHg.

### 2.3. Clinical Data Collection and Outcome Assessments

All clinical data were obtained from computerized medical record system. Demographic data, intraoperative data, and pre- and postoperative laboratory data were collected. Demographic data included age, sex, weight, height, body mass index (BMI), diabetes mellitus (DM), hypertension (HTN), cerebrovascular accident (CVA), smoking history, and the American Society of Anesthesiologists (ASA) physical status classification. Intraoperative data included tumor location, tumor invasion, lymph node invasion, distant metastasis, operation time, lowest mean blood pressure (MBP), laparoscopic surgery, and administered fluid volumes, such as crystalloid, colloid, transfused packed RBC, urine output, and diuretics. The laboratory data included preoperative white blood cell (WBC), hemoglobin, albumin, creatinine, estimated glomerular filter ratio (eGFR), neutrophil–lymphocyte ratio (NLR), platelet–lymphocyte ratio (PLR), red cell distribution width (RDW), and PNI. NLR was defined as the ratio between absolute neutrophil count to absolute lymphocyte count. PLR was determined as the ratio between absolute platelet counts to absolute lymphocyte count. PNI was calculated using the following equation: [10 × serum albumin (g dL^−1^)] + [0.005 × total lymphocyte count].

Serum creatinine (sCr) levels were checked daily from postoperative day 1 to postoperative day 7 to identify postoperative AKI by Kidney Disease Improving Global Outcomes classification (KDIGO): Increase in sCr by >1.5-fold from preoperative baseline within 7 days or increase in sCr ≥ 0.3 mg dL^−1^ within 48 h. The KDIGO classification is a newly introduced and rapidly adopted AKI definition for high sensitivity and early diagnosis [20]. Data on the length of hospital stay, postoperative ICU admission, prolonged ICU stay (more than 2 days), postoperative complications (cardiac and pulmonary problems, and bleeding,), 1-year mortality (calculated from the date of surgery to 1-year follow-up), and overall mortality (determined from the date of surgery to the last follow-up) were also collected.

### 2.4. Primary and Secondary Outcomes

The primary outcome was the incidence of postoperative AKI as defined by KDIGO criteria. The secondary outcomes were the length of hospital stay, postoperative ICU admission, prolonged ICU stay (>2 days), postoperative complications (cardiac and pulmonary problems, and bleeding), 1-year mortality, and overall mortality.

### 2.5. Statistical Analysis

Categorical data were analyzed using the chi-square test, and continuous data were evaluated by analysis of variance (ANOVA). Data are presented as mean and standard deviation (SD), median with interquartile range, or numbers with proportions, as appropriate. We used multivariate logistic regression analysis to determine independent predictors of AKI. We included all variables that showed statistical difference over increasing quartiles of PNI (standardized mean difference [SMD] >0.1), variables with *p* < 0.1 in univariate, and prior knowledge of important variables for AKI and mortality to the multivariable adjusted analysis. To assess the adjusted odd ratios (OR) of the risk factors of 1-year mortality, multivariable logistic regression analysis was also used. The cumulative survival rate among quartile groups was analyzed using the Kaplan–Meier method, and variations between curves were assessed using the log-rank test. We used two-tailed *p* values in tests of significance. All *p* values < 0.05 were considered statistically significant. Data manipulation and statistical analysis were performed using IBM SPSS Statistics for Windows, version 22.0 (IBM Corp., Armonk, NY, USA).

## 3. Results

Of the 3843 enrolled patients, 300 met the exclusion criteria. Finally, a total of 3543 patients were enrolled in this study. The included 3543 patients were classified into four groups by PNI quartile; Quartile Ⅰ (PNI ≤ 43.79, *n* = 885), Quartile Ⅱ (43.79 < PNI ≤ 47.79, *n* = 885), Quartile Ⅲ (47.79 < PNI ≤ 51.62, *n* = 887), Quartile Ⅳ (PNI > 51.62, *n* = 886) (Figure 1).

Table 1 shows demographic and perioperative variables of study population and SMD of each variable. The lower preoperative PNI groups tended to be older, more likely to be female, lower BMI, poor ASA class, and an increased chance of history of CVA and smoking than higher preoperative PNI groups (Table 1). The groups with lower preoperative PNI tended to have lower levels of hemoglobin, albumin, and creatinine, and higher levels of NLR, PLR, and RDW (Table 1). Operation time tended to be more in the lower preoperative PNI groups, and total fluid, crystalloid, and RBC transfusion rates tended to be higher in the lower preoperative PNI groups (Table 1). In the lower preoperative PNI groups, lowest MBP tended to be lower, and laparoscopic surgery was performed more in the higher preoperative PNI group (Table 1).

### 3.1. Primary Outcomes

The postoperative AKI incidence by PNI quartile is shown in Table 1. The AKI incidence rates were 13.4% (119/885) in the Q1 group, 9.2% (81/885) in the Q2 group, 9.4% (83/887) in the Q3 group, and 8.8% (78/886) in the Q4 group.

### 3.2. Secondary Outcomes

The lower preoperative PNI groups tended to have longer hospital days (9.45 days in Q1, 8.01 days in Q2, 7.69 days in Q3, 7.41 days in Q4), more ICU admissions (6.1% in Q1, 2.8% in Q2, 2.5% in Q3, 2.0% in Q4), and ICU stay ≥2 days (2.3% in Q1, 0.5% in Q2, 0.7% in Q3, 0.7% in Q4) (Table 1). The lower preoperative PNI groups tended to have lower 1-year mortality (1.9% in Q1, 0.3% in Q2, 0.2% in Q3, 0.3% in Q4), and overall mortality (2.9% in Q1, 0.3% in Q2, 0.5% in Q3, 0.7% in Q4) (Table 1). Mortality was followed for 3.14 (2.24–3.93) years, and the last follow-up was 26 March 2013.

In the multivariate analysis, low preoperative PNI was significantly associated with an increased risk of postoperative AKI (OR: 0.96, 95% CI: 0.93–0.99, *p* = 0.003). Additionally, male gender (OR 1.60, 95%CI 1.19–2.14, *p* < 0.001), BMI (OR 1.06, 95%CI 1.02–1.11, *p* = 0.003), DM (OR 1.48, 95%CI 1.10–1.98, *p* = 0.009), HTN (OR 1.46, 95%CI 1.12–1.91, *p* = 0.006), white blood cell (OR 1.10, 95%CI 1.02–1.19, *p* = 0.016), and creatinine (OR 0.05, 95%CI 0.02–0.13, *p* < 0.001) were significantly associated with postoperative AKI (Table 2).

In the multivariate logistic regression analysis of 1-year mortality, low preoperative PNI (OR: 0.92, 95% CI: 0.86–0.98, *p* = 0.011), DM (OR: 3.91, 95% CI: 1.60–9.60, *p* = 0.003), and smoking (OR: 4.12, 95% CI: 1.28–13.31, *p* = 0.018) were significantly associated with higher 1-year mortality (Table 3).

Preoperative PNI quartile 1 group was significantly associated with the incidence of postoperative AKI (OR 1.52, 95%CI 1.18–1.95, *p* = 0.001), hospital days ≥ 14 days (OR 2.16, 95%CI 1.55–2.99, *p* < 0.001), ICU stay ≥ 2 days (OR 2.14, 95%CI 1.01–4.53, *p* = 0.046), 1-year mortality (OR 3.83, 95%CI 1.55–9.49, *p* = 0.004), and overall mortality (OR 3.81, 95%CI 1.86–7.79, *p* < 0.001) even after adjusting for other potentially confounding variables (Table 4).

Figure 2 shows the Kaplan–Meier curve according to preoperative PNI level quartiles (log-rank test; *p* < 0.001). The 1-year and overall survival were significantly different between the PNI level of quartile 1 group and PNI level of quartile 2, 3, and 4 groups. Appendix A shows the Kaplan–Meier curve of postoperative AKI as defined by KDIGO criteria.

## 4. Discussion

Our study demonstrated that low preoperative PNI was significantly associated with postoperative AKI in patients who underwent colorectal cancer surgery. In addition, low preoperative PNI was significantly associated with surgical outcomes, such as increased hospital days, prolonged (more than 2 days) ICU stay, and reduced 1-year and overall mortality. This suggests that the preoperative PNI might be a prognostic factor for postoperative AKI incidence and surgical outcome in colorectal cancer patients.

PNI comprises of assessment of albumin, a major component of plasma protein, and lymphocytes, which are important cells in immunity, indicating patient’s nutritional and immune status [21]. There have been many studies reporting that low PNI is significantly associated with postoperative complications, including surgical site infection, anastomotic leakage, bleeding, cardiopulmonary failure, and pulmonary embolism [22,23,24,25]. However, only few studies have reported the association between PNI and AKI in surgical patients. Min and colleagues suggested that the modified PNI predicted postoperative AKI within 1-week better than the conventional model of end-stage liver disease score (MELD) in patients receiving living donor liver transplantation and recommended a cutoff-value of 8.7 (area under curve 0.823, sensitivity 72.2%, specificity 83.2%) [26]. Dolapoglu and colleagues demonstrated that an association between PNI and AKI in patients with normal serum creatinine levels who underwent coronary artery bypass grafting [27].

Our study has clinical significance as it is the first study to investigate the association between preoperative PNI and postoperative AKI in patients undergoing colorectal cancer surgery. The results of the current study are consistent with those of the other two studies mentioned above, which reported the association between PNI and AKI. In our study, after dividing the PNI level into quartiles, the incidence of postoperative AKI decreased significantly as the PNI quartile increased sequentially, which was statistically significant. In particular, the incidence of AKI increased significantly in patients with PNI level of quartile 1. This finding suggests that low preoperative PNI is significantly associated with postoperative AKI.

Following multivariate logistic regression analysis, male gender, DM, HTN, BMI, and preoperative PNI were found to be significantly associated with postoperative AKI. Male gender has been reported to be associated with an increased risk of hospital-associated AKI [28]. One recent animal study found that female sex prevents the development of AKI, and sex hormones might be involved in this protection against AKI [29].

DM and HTN have been reported as risk factors of AKI in surgical patients [30,31]. DM is related to post-ischemic micro-vasculopathy and interstitial inflammation [31]. Furthermore, DM and HTN were known to be associated with conditions that can cause AKI even in the absence of chronic kidney disease [30]. Previous studies on the marginal association between BMI and AKI have reported potential confounding factors, including DM and HTN [32].

Recent studies have reported that biological markers, neutrophil-to-lymphocyte ratio (NLR) and platelet-to-lymphocyte ratio (PLR), are significantly associated with AKI and renal replacement therapy in surgical patients [33,34]. However, in our study, NLR and PLR were not significantly associated with postoperative AKI in colorectal cancer patients. In the studies by Bi et al. and Kim et al., 8.38 and 11.7 were proposed as cutoff values of NLR predicting AKI [33,34]. However, in our study, patients with NLR ≥ 8.38 were 1.0% (37/3543), and with NLR ≥ 11.7 were 0.5% (18/3543), and few patients had high preoperative NLR levels, so, there might be no association between NLR and AKI.

In the present study, preoperative PNI was the only biological marker associated with postoperative AKI, and this association between AKI and PNI seems to be due to the characteristics of PNI that reflect the function of albumin and lymphocytes. Albumin is responsible for maintaining oncotic pressure, microvascular permeability, and acid–base balance, and preventing platelet aggregation [35]. These functions of albumin might be associated with the reno-protective properties. Potent renal vasodilation caused by serum albumin and nitric oxide due to the formation of S-nitroso-albumin has been reported to increase renal perfusion in animal models [36]. Serum albumin plays an important role in proximal tubular homeostasis, and hypoalbuminemia has been reported as a strong risk factor for postoperative AKI in both cardiac and non-cardiac surgery [37,38]. Additionally, it has also been reported that albumin may mitigate the effects of nephrotoxic drugs [39] and stimulate renal cell proliferation through a pathway mediated by phosphatidyl inositide 3-kinase [40]. Lymphocytes are also known to play an important role in the process of AKI initiation, proliferation, and recovery [41], and one study showed a significant association between preoperative lymphocytopenia and postoperative AKI in cardiac surgery [42]. Additionally, recently newly identified renal T lymphocytes have been reported to have complex functions, such as potentially playing an anti-inflammatory role in AKI [43].

Our study demonstrated that PNI level of quartile 1 (PNI ≤ 43.79) was significantly associated with postoperative AKI and surgical outcomes, such as hospital day, prolonged (more than 2 days) ICU stay, and 1-year and overall mortality. In Korea, all citizens are required to sign up for a national health insurance system, and when a patient dies, the death report is included in the database. In addition, patient survival data are automatically updated in the patient medical records in our center, accompanied by loss of health insurance eligibility. In recent studies, PNI < 45 was proposed as a cut off value that was significantly associated with postoperative complications or survival rate [21,22,23,24,44,45], which is consistent with our results. Therefore, it seems that a preoperative PNI level of less than 45 may be associated with a surgical prognosis.

There are several limitations in our study. First, our study is retrospective, and it is possible that confounding factors that have not been considered might have caused potential errors. A total of 193 patients (5.2%) with missing data such as hemoglobin (77 patients), eGFR (48 patients), RDW (32 patients) were excluded from the study, which may have affected surgical outcome. However, their characteristics were not significantly different from the patients who included in the study (Appendix A) and because of their small number, we do not think it would have had a significant impact on our findings. In our study, stoma creation rates and surgical site leakage were not analyzed. Therefore, careful interpretation of some surgical results is required. Second, since our data were collected from a single medical center, the results might have been biased due to homogeneous groups; thus, further study of heterogeneous groups is needed. Third, to date, there is no exact agreement on the cutoff value of PNI, and there are slight differences depending on the study. More well-designed studies are needed for accurate validation of preoperative PNI cutoff value that predict complications and survival rates.

## 5. Conclusions

In conclusion, preoperative PNI was associated with postoperative AKI and surgical outcomes in patients who underwent colorectal cancer surgery. These results suggest that preoperative PNI can provide useful information about postoperative AKI and surgical prognosis in colorectal cancer.

## Figures and Tables

**Figure 1 nutrients-13-01604-f001:**
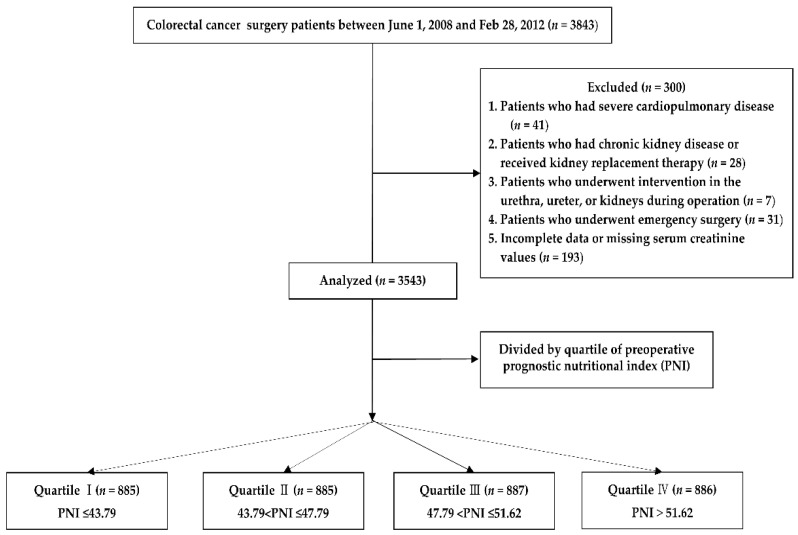
Study flow chart.

**Figure 2 nutrients-13-01604-f002:**
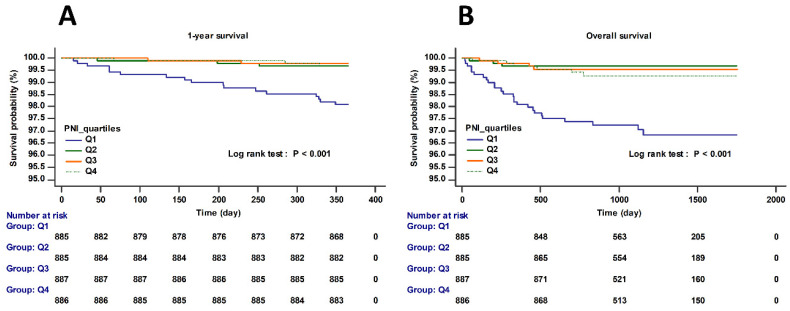
Kaplan–Meier survival curve for (**A**) 1-year and (**B**) overall survival (log-rank test; *p* < 0.001).

**Table 1 nutrients-13-01604-t001:** Demographic, perioperative variables, and surgical outcomes of study population.

	Prognostic Nutritional Index				
	Quartile Ⅰ (*n* = 885)	Quartile Ⅱ (*n* = 885)	Quartile Ⅲ (*n* = 887)	Quartile Ⅳ (*n* = 886)	SMD
Preoperative variables					
Age; year	62.60 ± 11.83	60.73 ± 10.64	59.08 ± 10.32	56.69 ± 10.92	0.293
Sex; male	512 (57.9)	528 (59.7)	551 (62.1)	585 (66.0)	0.093
Weight; kg	59.56 ± 9.84	62.09 ± 9.87	64.43 ± 10.33	65.36 ± 10.84	0.322
Height; cm	1.61 ± 0.08	1.62 ± 0.09	1.63 ± 0.08	1.63 ± 0.09	0.143
BMI; kg m^−2^	22.90 ± 3.17	23.71 ± 3.01	24.25 ± 2.92	24.42 ± 3.06	0.276
DM	136 (15.4)	119 (13.4)	134 (15.1)	134 (15.1)	0.027
HTN	288 (32.5)	270 (30.5)	296 (33.4)	320 (36.1)	0.063
CVA	26 (2.9)	23 (2.6)	11 (1.2)	11 (1.2)	0.076
Smoking	513 (58.0)	504 (56.9)	481 (54.2)	439 (49.5)	0.144
ASA status					0.197
ASA 1	148 (16.7)	210 (23.7)	227 (25.6)	255 (28.8)	
ASA 2	694 (78.4)	660 (74.6)	650 (73.3)	624 (70.4)	
ASA 3	43 (4.9)	15 (1.7)	10 (1.1)	7 (0.8)	
Laboratory variables					
White blood cell	5.85 ± 2.32	5.69 ± 1.85	6.03 ± 1.69	6.95 ± 1.72	0.357
Hemoglobin	11.20 ± 1.80	12.21 ± 1.75	12.63 ± 1.77	13.12 ± 1.64	0.591
Albumin; g dL^−1^	3.31 ± 0.40	3.80 ± 0.26	4.02 ± 0.25	4.24 ± 0.26	1.626
Creatinine; mg dL^−1^	0.74 ± 0.18	0.79 ± 0.18	0.81 ± 0.18	0.83 ± 0.17	0.284
eGFR; mL.min^−1^.1.73 m^−2^	75.52 ± 13.63	75.96 ± 13.18	74.56 ± 12.98	74.47 ± 12.95	0.069
NLR	3.59 ± 3.21	2.42 ± 1.37	1.95 ± 0.91	1.59 ± 0.69	0.607
PLR	226.50 ± 124.76	165.69 ± 62.47	135.97 ± 47.84	107.72 ± 36.53	0.487
RDW	15.20 ± 3.37	14.19 ± 2.63	13.55 ± 2.13	13.15 ± 1.79	0.436
PNI	39.37 ± 3.88	45.87 ± 1.17	49.69 ± 1.10	54.79 ± 2.73	3.421
Intraoperative variables					
Tumor location					0.025
colon	624 (70.51)	640 (72.32)	633 (71.36)	623 (70.32)	
rectum	261 (29.49)	245 (27.68)	254 (28.64)	263 (29.68)	
Tumor invasion					0.288
Tis, T1, T2	129 (14.58)	181 (20.45)	233 (26.27)	329 (37.13)	
T3, T4	756 (85.42)	704 (79.55)	654 (73.73)	557 (62.87)	
Lymph node invasion					0.291
N0	310 (35.03)	408 (46.10)	481 (54.23)	538 (60.72)	
N1, N2	575 (64.97)	477 (53.90)	406 (45.77)	348 (39.28)	
Distant metastasis					0.098
M0	765 (86.44)	794 (89.72)	799 (90.08)	818 (92.33)	
M1	120 (13.56)	91 (10.28)	88 (9.92)	68 (7.67)	
Operation time; min	180.05 ± 66.45	169.48 ± 57.67	167.83 ± 60.13	166.86 ± 58.35	0.111
Lowest MBP; mmHg	69.33 ± 9.05	70.91 ± 8.47	71.65 ± 9.02	72.27 ± 8.80	0.180
Laparoscopic surgery	159 (18.0)	217 (24.5)	260 (29.3)	338 (38.1)	0.247
Total fluids; mL kg^−1^	30.93 ± 15.72	27.41 ± 12.51	26.04 ± 10.36	25.74 ± 9.86	0.218
Crystalloid; mL kg^−1^ h^−1^	24.45 ± 14.05	21.44 ± 11.57	19.85 ± 9.55	19.69 ± 9.03	0.226
Colloid; mL kg^−1^ h^−1^	6.42 ± 5.33	5.96 ± 4.81	6.19 ± 4.25	6.05 ± 4.25	0.054
Colloid use	616 (69.6)	604 (68.2)	671 (75.6)	660 (74.5)	0.101
RBC transfusion	51 (5.8)	20 (2.3)	18 (2.0)	10 (1.1)	0.134
Unit of infused RBC	0.11 ± 0.53	0.04 ± 0.33	0.05 ± 0.38	0.03 ± 0.36	0.094
Urine output; mL kg^−1^ h^−1^	1.88 ± 1.74	1.88 ± 1.79	1.84 ± 1.44	2.00 ± 1.59	0.052
Diuretics	8 (0.9)	2 (0.2)	4 (0.5)	4 (0.5)	0.046
Surgical outcomes					
AKI	119 (13.4)	81 (9.2)	83 (9.4)	78 (8.8)	0.075
Hospital days	9.45 ± 9.28	8.01 ± 5.51	7.69 ± 4.37	7.41 ± 5.69	0.154
ICU admission	54 (6.1)	25 (2.8)	22 (2.5)	18 (2.0)	0.117
ICU stay (≥2 days)	20 (2.3)	4 (0.5)	6 (0.7)	6 (0.7)	0.153
Postoperative complication	10 (1.1)	9 (1.0)	8 (0.9)	6 (0.7)	
Cardiac	1 (0.1)	2 (0.2)	1 (0.1)	2 (0.2)	
Pulmonary	4 (0.5)	6 (0.7)	5 (0.6)	2 (0.2)	
Bleeding	5 (0.6)	1 (0.1)	2 (0.2)	2 (0.2)	
1-year mortality	17 (1.9)	3 (0.3)	2 (0.2)	3 (0.3)	0.085
Overall mortality	26 (2.9)	3 (0.3)	4 (0.5)	6 (0.7)	0.111

Quartile Ⅰ, first quarter group (PNI ≤ 43.79); Quartile Ⅱ, second quarter group (PNI 43.79–47.79); Quartile Ⅲ, third quarter group (PNI 47.79–51.62); Quartile Ⅳ, fourth quarter group (PNI ≥51.62); BMI, body mass index; DM, diabetes mellitus; HTN, hypertension; CVA, cerebrovascular accident; ASA, American Society of Anesthesiologists classification; eGFR, estimated glomerular filtration rate; NLR, neutrophil lymphocyte ratio; PLR, platelet lymphocyte ratio; RDW, red cell distribution width; PNI, prognostic nutritional index; MBP, mean blood pressure; RBC, red blood cell; AKI, acute kidney injury; ICU, intensive care unit; SD, standard deviation; SMD, standardized mean difference. Values are expressed as the mean (SD), median (interquartile range), or n (proportion).

**Table 2 nutrients-13-01604-t002:** Univariate and multivariate logistic regression analysis of AKI.

	Univariate			Multivariate		
	OR	95% CI	*p*-Value	OR	95% CI	*p*-Value
PNI	0.97	0.96–0.99	0.003	0.96	0.93–0.99	0.003
Age	1.01	1.00–1.02	0.018	1.00	0.99–1.02	0.465
Sex (male)	1.65	1.30–1.21	<0.001	1.60	1.19–2.14	<0.001
BMI	1.04	1.00–1.07	0.042	1.06	1.02–1.11	0.003
DM	1.83	1.40–2.38	<0.001	1.48	1.10–1.98	0.009
HTN	1.50	1.20–1.87	<0.001	1.46	1.12–1.91	0.006
CVA	1.29	0.63–2.61	0.485			
Smoking	1.43	1.13–1.80	0.003	1.09	0.81–1.45	0.573
ASA			0.010			0.130
ASA status 1	1.00			1.00		
ASA status 2	1.20	0.91–1.56	0.195	0.80	0.58–1.11	0.189
ASA status 3	2.59	1.40–4.78	0.002	1.36	0.68–2.73	0.388
Tumor location						
colon	1.00			1.00		
rectum	0.90	0.70–1.14	0.376			
Tumor invasion						
Tis, T1, T2	1.00			1.00		
T3, T4	1.21	0.93–1.58	0.153	1.12	0.81–1.54	0.497
Lymph node invasion						
N0	1.00			1.00		
N1, N2	1.17	0.94–1.46	0.152	1.02	0.79–1.33	0.862
Distant metastasis						
M0	1.00			1.00		
M1	1.28	0.91–1.79	0.157	1.09	0.74–1.58	0.672
Operation time; min	1.00	1.00–1.00	0.333	1.00	1.00–1.00	0.798
Laparoscopic surgery	0.98	0.77–1.25	0.878	1.05	0.80–1.39	0.712
Lowest MBP; mmHg	1.00	0.99–1.02	0.625	1.00	0.99–1.02	0.496
Total fluids; mL kg^−1^	1.00	0.99–1.01	0.565	1.00	0.99–1.02	0.748
Crystalloid; mL kg^−1^	1.00	1.00–1.01	0.289			
Colloid; mL kg^−1^	0.99	0.96–1.01	0.244			
Synthetic Colloid use	0.91	0.72–1.16	0.464	0.98	0.75–1.27	0.862
Urine output; mL kg^−1^ h^−1^	1.03	0.97–1.10	0.372	1.06	0.99–1.13	0.124
Diuretics	1.77	0.51–6.14	0.369	1.57	0.35–5.19	0.497
RBC transfusion	1.35	0.74–2.44	0.327	1.04	0.51–2.11	0.922
White blood cell	1.04	0.98–1.09	0.179	1.10	1.02–1.19	0.016
Hemoglobin	0.98	0.93–1.04	0.560	0.96	0.89–1.04	0.316
Albumin; g dL^−1^	0.55	0.44–0.69	<0.001			
Creatinine	0.38	0.21–0.71	0.002	0.05	0.02–0.13	<0.001
eGFR	1.00	0.99–1.01	0.608			
NLR	0.98	0.93–1.05	0.594	0.90	0.78–1.03	0.118
PLR	1.00	0.99–1.01	0.302	0.98	0.94–1.02	0.337
RDW	1.01	0.97–1.05	0.755	0.99	0.94–1.03	0.579

AKI, acute kidney injury; SD, standard deviation; OR, odds ratio; CI, confidence interval; PNI, prognostic nutritional index; BMI, body mass index; DM, diabetes mellitus; HTN, hypertension; CVA, cerebrovascular accident; ASA, American Society of Anesthesiologists classification; eGFR, estimated glomerular filtration rate; NLR, neutrophil lymphocyte ratio; PLR, platelet lymphocyte ratio; RDW, red cell distribution width; MBP, mean blood pressure; RBC, red blood cell. Values are expressed as the mean (SD), median (interquartile range), or n (proportion).

**Table 3 nutrients-13-01604-t003:** Univariate and multivariate logistic regression analysis of 1-year mortality.

	Univariate			Multivariate		
	OR	95% CI	*p*-Value	OR	95% CI	*p*-Value
PNI	0.87	0.82–0.92	<0.001	0.92	0.86–0.98	0.011
Age	1.05	1.01–1.10	0.008	1.04	1.00–1.09	0.054
Sex (male)	1.12	0.49–2.54	0.790	1.50	0.41–5.50	0.545
BMI	0.90	0.79–1.02	0.111	1.02	0.89–1.18	0.760
DM	3.91	1.75–8.74	<0.001	3.91	1.60–9.60	0.003
HTN	1.87	0.85–4.11	0.119	1.33	0.49–3.62	0.577
CVA	2.05	0.27–15.38	0.484			
Smoking	2.96	1.3–6.72	0.009	4.12	1.28–13.31	0.018
ASA			<0.001			0.130
ASA status 1	1.00			1.00		
ASA status 2	1.82	0.53–6.21	0.342	0.56	0.14–2.28	0.418
ASA status 3	19.93	4.67–85.13	<0.001	2.93	0.52–16.41	0.221
Tumor location						
colon	1.00			1.00		
rectum	1.39	0.62–3.13	0.430			
Tumor invasion						
Tis, T1, T2	1.00			1.00		
T3, T4	7.49	1.02–54.90	0.049	3.58	0.38–33.67	0.265
Lymph node invasion						
N0	1.00			1.00		
N1, N2	2.89	1.15–7.24	0.025	1.81	0.69–4.78	0.230
Distant metastasis						
M0	1.00			1.00		
M1	2.94	1.17–7.38	0.022	1.67	0.56–4.99	0.358
Operation time; min	1.00	1.00–1.01	0.022	1.01	1.00–1.01	0.116
Laparoscopic surgery	0.11	0.01–0.81	0.030	0.15	0.02–1.16	0.070
Lowest MBP; mmHg	0.95	0.90–0.99	0.027	1.00	0.94–1.05	0.853
Total fluids; mL kg^−1^	1.02	1.00–1.04	0.030	1.27	0.90–1.80	0.178
Crystalloid; mL kg^−1^	1.03	1.01–1.05	0.005	0.80	0.56–1.13	0.208
Colloid; mL kg^−1^	0.94	0.86–1.03	0.167	0.72	0.49–1.04	0.080
Synthetic Colloid use	0.42	0.19–0.92	0.030	0.40	0.16–1.02	0.054
Urine output; mL kg^−1^ h^−1^	0.85	0.63–1.16	0.305			
Diuretics	*		0.992			
RBC transfusion	3.07	0.71–13.19	0.132	0.59	0.09–3.93	0.593
White blood cell	1.13	0.96–1.34	0.142	1.06	0.87–1.30	0.547
Hemoglobin	0.7	0.58–0.85	<0.001	0.92	0.70–1.20	0.534
Albumin; g dL^−1^	0.19	0.10–0.36	<0.001			
Creatinine	0.11	0.01–1.05	0.05503	0.15	0.01–3.27	0.228
eGFR	1.00	0.97–1.03	0.930			
NLR	1.10	1.02–1.18	0.010	0.98	0.73–1.31	0.888
PLR	1.02	1.00–1.05	0.003	1.00	0.92–1.09	0.952
RDW	1.19	1.09–1.31	<0.001	1.08	0.95–1.22	0.256

* cannot be calculated because the number of samples is small. AKI, acute kidney injury; SD, standard deviation; OR, odds ratio; CI, confidence interval; PNI, prognostic nutritional index; BMI, body mass index; DM, diabetes mellitus; HTN, hypertension; CVA, cerebrovascular accident; ASA, American Society of Anesthesiologists classification; eGFR, estimated glomerular filtration rate; NLR, neutrophil lymphocyte ratio; PLR, platelet lymphocyte ratio; RDW, red cell distribution width; MBP, mean blood pressure; RBC, red blood cell. Values are expressed as the mean (SD), or n (proportion).

**Table 4 nutrients-13-01604-t004:** AKI incidence and surgical outcomes adjusted by PNI.

	Univariate		Multivariate *	
	OR (95% CI)	*p*-Value	OR (95% CI)	*p*-Value
AKI				
Quartile Ⅰ	1.55 (1.23–1.96)	<0.001	1.52 (1.18–1.95)	0.001
Quartile Ⅱ, Ⅲ, Ⅳ	1.00		1.00	
Postoperative complication				
Quartile Ⅰ	1.51 (0.73–3.12)	0.269	0.86 (0.38–1.94)	0.721
Quartile Ⅱ, Ⅲ, Ⅳ	1.00		1.00	
Hospital days (≥14 days)				
Quartile Ⅰ	2.86 (2.14–3.82)	<0.001	2.16 (1.55–2.99)	<0.001
Quartile Ⅱ, Ⅲ, Ⅳ	1.00		1.00	
ICU admission				
Quartile Ⅰ	2.59 (1.79–3.75)	<0.001	1.26 (0.79–2.01)	0.327
Quartile Ⅱ, Ⅲ, Ⅳ	1.00		1.00	
ICU stay (≥2 days)				
Quartile Ⅰ	3.82 (1.97–7.40)	<0.001	2.14 (1.01–4.53)	0.046
Quartile Ⅱ, Ⅲ, Ⅳ	1.00		1.00	
1-year mortality				
Quartile Ⅰ	6.49 (2.79–15.09)	<0.001	3.83 (1.55–9.49)	0.004
Quartile Ⅱ, Ⅲ, Ⅳ	1.00		1.00	
Overall mortality				
Quartile Ⅰ	6.16 (3.15–12.04)	<0.001	3.81 (1.86–7.79)	<0.001
Quartile Ⅱ, Ⅲ, Ⅳ	1.00		1.00	

* Adjusted for age, sex, ASA, DM, CVA, tumor invasion, lymph node invasion, distant metastasis, operation time, lowest MBP, total fluids, and RBC transfusion. Quartile Ⅰ, first quarter group (PNI ≤ 43.79); Quartile Ⅱ, second quarter group (PNI 43.79–47.79); Quartile Ⅲ, third quarter group (PNI 47.79–51.62); Quartile Ⅳ, fourth quarter group (PNI ≥ 51.62); AKI, acute kidney injury; SD, standard deviation; OR, odds ratio; CI, confidence interval; PNI, prognostic nutritional index; DM, diabetes mellitus; CVA, cerebrovascular accident; ASA, American Society of Anesthesiologists classification; MBP, mean blood pressure; RBC, red blood cell. Values are expressed as the mean (SD), median (interquartile range), or n (proportion).

## Data Availability

The dataset used and/or analyzed during the current study is available from the corresponding author on reasonable request.

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
