# Peer review of "Association of Preoperative Prognostic Nutritional Index and Postoperative Acute Kidney Injury in Patients with Colorectal Cancer Surgery"

_nutrients, 2021, doi:10.3390/nu13051604_

Round 1

Reviewer 1 Report

Overall, this was a well-written manuscript looking at a way to more accurately assess the preoperative nutritional and immune status of patients and how that might impact postoperative outcomes, namely acute kidney injury. Please see below for specific comments.

 Title/Abstract

  1. Would suggest revising sentence to “The Q1 group had poor surgical outcomes, such as higher rates of ICU admission…and lower rates of 1-year survival…”
  2. Don’t understand the sentence at line 23-25 beginning “The postoperative AKI and surgical outcomes…”
  3. What is your definition of “poor surgical outcomes”?
  4. Please provide definition of postoperative AKI in abstract.

Introduction

  1. Once referring to acute kidney injury as AKI, refer to as AKI throughout.
  2. What kinds of complications were analyzed?
  3. Would spend a little more time describing PNI, as in how it works. Authors describe separating PNI into quartiles, presumably lowest quartile is “worst”, but would explain this.

Materials and Methods

  1. Were all of these patients operated on at the Asan Medical Center?
  2. Did you separate out colon and rectal cases and laparoscopic or open cases?
  3. There is a large amount of detail provided regarding intraoperative anesthesia care, though I am not entirely sure so much detail is required. I think that the intraoperative values for hemodynamics would be of more utility.
  4. Again, please expound on what was included in postoperative complications.

Results

  1. For 1-year survival and overall survival, is this actually referring to mortality at 1 year and overall? Because as reported now, it implies only 1.9% of patients were still alive in the QI group and so on. Would either report the inverse of those numbers, or change the variables to 1-year mortality, overall mortality.
  2. What is the definition of overall survival? Is this a longer time period?
  3. In the multivariate analysis, was PNI analyzed as a continuous variable? Would recommend further describing the association as well, such as lower PNI was associated with an increased risk of postoperative AKI.
  4. Again, the regression for 1-year survival is confusing based on perceived definition of 1-year survival. Would also further describe these associations. PNI not just associated with survival, but how? Lower or higher PNI with lower or higher survival.
  5. For the regression reported in Table 4, what other “potentially confounding variables” were controlled for?
  6. For the Kaplan Meier, how long were these patients followed for? What is the breakdown for overall follow up time per quartile?
  7. Is more granular data available regarding postoperative complications? I think that is likely more meaningful than ICU days or admission. Certainly specific complications of great interest to colorectal surgeons would include things such as surgical site infections, ileus, postoperative leak, return to operating room, 30-day readmission. These would also make sense to be associated with poor nutritional status.
  8. Is there information on stoma creation rates? Rates of rectal vs colon cancer? LAR vs APR?

Discussion

  1. As these were cancer patients, data regarding clinical and/or pathologic stage seems important as this could be a major confounding factor and clearly higher stage patients are probably more likely to be more nutritionally deplete as well as more likely to die due to disease.

Conclusion

Author Response

Responses to Reviewer #1

 Title/Abstract

  1. Would suggest revising sentence to “The Q1 group had poor surgical outcomes, such as higher rates of ICU admission…and lower rates of 1-year survival…”

Response: Thank you for the thoughtful suggestion. The sentence was clarified, as follows, “The Q1 group had poor surgical outcomes, such as higher rates of ICU admission (adjusted OR: 3.13, 95% CI: 1.82–5.39, p<0.001) and higher 1-year mortality (adjusted OR: 5.77, 95% CI: 1.68–19.74, p=0.005).” (page 1, lines 23-25).

  1. Don’t understand the sentence at line 23-25 beginning “The postoperative AKI and surgical outcomes…”

Response: Thank you for pointing out the lack of clarity. We intended to say that low preoperative PNI, especially in the Q1 (≤43.79), was significantly associated with poor surgical outcomes such as length of hospital stay, postoperative ICU admission, and mortality. The text was revised as follows: “In conclusion, low preoperative PNI levels, especially in the Q1 (≤43.79), were significantly associated with postoperative AKI and surgical outcomes, such as hospital stay, postoperative ICU admission, and mortality.” (page 1, lines 26-28).

  1. What is your definition of “poor surgical outcomes”?

Response: We appreciate this relevant question. We compared postoperative outcomes according to PNI level by assessing the length of stay, postoperative ICU admission, postoperative complications, 1-year mortality, and overall mortality. In order to avoid ambiguity, we referred to poor surgical outcomes as surgical outcomes throughout the text.

  1. Please provide definition of postoperative AKI in abstract.

Response: Postoperative AKI was defined according to KDIGO criteria: “AKI was defined according to Kidney Disease Improving Global Outcomes classification (KDIGO) criteria.” (page 1, lines 17-18).

 Introduction

  1. Once referring to acute kidney injury as AKI, refer to as AKI throughout.

Response: Thank you for the recommendation. We referred to acute kidney injury as AKI throughout the text: “PNI is an indicator of patient survival and complications in acute and chronic diseases, such as AKI, heart failure, and various cancers.” (page 2, line 45).

  1. What kinds of complications were analyzed?

Response: Thank you for asking. In total, 37 patients (2.5%) had postoperative complications, including cardiac (N=6) and pulmonary complications (N=17), bleeding (N=10), and leakage (N=4). The text was improved as follows: “Data on the length of hospital stay, postoperative ICU admission, prolonged ICU stay (more than 2 days), postoperative complications (cardiac and pulmonary problems, bleeding, and leakage), 1-year mortality (calculated from the date of surgery to 1-year follow-up), and overall mortality (determined from the date of surgery to the last follow-up) were also collected” (Table 1, page 3, lines 98-102).

  1. Would spend a little more time describing PNI, as in how it works. Authors describe separating PNI into quartiles, presumably lowest quartile is “worst”, but would explain this.

Response: We agree with you. More emphasis was given to PNI as follows: “Malnutrition has a significant impact on patient prognosis by reducing immune function, and lymphocyte count is an indicator of cell-mediated immunity and plays an important role in cancer prevention [8,9]. Therefore, more attention has been paid to the impact of nutritional and immune status on the prognosis of various cancers.” (page 1, lines 40-43).

The following studies analyzed PNI according to quartiles:

1. Hu, Y.; Cao, Q.; Wang, H.; Yang, Y.; Xiong, Y.; Li, X.; Zhou, Q. Prognostic nutritional index predicts acute kidney injury and mortality of patients in the coronary care unit. Exp Ther Med 2021;21:123.

2. Okada S, Shimada J, Kato D, Tsunezuka H, Teramukai S, Inoue M. Clinical significance of prognostic nutritional index after surgical treatment in lung cancer. Ann Thorac Surg. 2017;104:296-302.

3. Yang L, Yu W, Pan W, et al. A clinical epidemiological analysis of prognostic nutritional index associated with diabetic retinopathy. Diabetes Metab Syndr Obes. 2021;14:839-846.

The analysis of PNI quartiles is not significantly affected by outliers and sample size and can identify trends in other variables.

Materials and Methods

  1. Were all of these patients operated on at the Asan Medical Center?

Response: Asan Medical Center is the largest tertiary hospital in Korea, with 2,715 beds, and performs approximately 70,000 surgeries annually, including more than 1,500 colorectal cancer surgeries. The present study was conducted from June 2008 to February 2012 (https://eng.amc.seoul.kr/gb/lang/specialities/departments.do?hpCd=D095).

    2. Did you separate out colon and rectal cases and laparoscopic or open cases?

Response: We separated colon and rectal cancer cases because tumor location can greatly influence surgical outcomes (Table 1). There was no significant difference in tumor location according to PNI quartiles. The text was clarified as follows: “Intraoperative data included tumor location, tumor invasion, lymph node invasion, dis-tant metastasis, operation time, lowest mean blood pressure (MBP), laparoscopic surgery, and administered fluid volumes, such as crystalloid, colloid, transfused packed RBC, urine output, and diuretics.” (page 2, line 83).

  1. There is a large amount of detail provided regarding intraoperative anesthesia care, though I am not entirely sure so much detail is required. I think that the intraoperative values for hemodynamics would be of more utility.

Response: We agree with you. We shortened the description of anesthetic techniques in Section 2.2 and added more information and references on the surgical methods that might affect hemodynamics, as follows: “Regardless of the surgical approach, all patients underwent general anesthesia (open vs. laparoscopy). All surgeries were performed by a qualified and experienced surgical team. Conventional and laparoscopic surgeries were performed using standard protocols [19]. For laparoscopic surgery, pneumoperitoneum with carbon dioxide at 10–15 mmHg was established. Crystalloid and colloid solutions were administered during surgery. The total volume of infused synthetic colloids did not exceed 20 mL kg-1. During surgery, packed red blood cells (RBC) were transfused if plasma hemoglobin level was less than 8 g dL-1, and inotropes or vasopressors were administered if the mean blood pressure was less than 65 mmHg.” (page 2, lines 67-76).

  1. Again, please expound on what was included in postoperative complications.

Response: Thank you for the inquiry. Postoperative complications included cardiac and pulmonary complications, bleeding, and surgical site leakage. The sentence was revised as follows: “Secondary outcomes were the length of hospital stay, postoperative ICU admission, prolonged ICU stay (>2 days), postoperative complications (cardiac and pulmonary problems, bleeding, and surgical site leakage), 1-year mortality, and overall mortality according to preoperative PNI quartiles.” (page 3, lines 105-108).

 Results

        1. For 1-year survival and overall survival, is this actually referring to mortality at 1 year and overall? Because as reported now, it implies only 1.9% of patients were still alive in the QI group and so on. Would either report the inverse of those numbers, or change the variables to 1-year mortality, overall mortality.

Response: We apologize for the confusion. The word “survival” was changed to “mortality” throughout the manuscript.

         2. What is the definition of overall survival? Is this a longer time period?

Response: We changed the word “survival” to “mortality” throughout the manuscript. Overall mortality was calculated from the date of surgery to the date of the last follow-up. The sentence was improved as follows: “Data on the length of hospital stay, postoperative ICU admission, prolonged ICU stay (more than 2 days), postoperative complications, 1-year mortality (calculated from the date of surgery to 1-year follow-up), and overall mortality (determined from the date of surgery to the date of the last follow-up) were also collected.” (page 3, lines 98-102).

         3. In the multivariate analysis, was PNI analyzed as a continuous variable? Would recommend further describing the association as well, such as lower PNI was associated with an increased risk of postoperative AKI.

Response: Thank you for the suggestion. PNI was considered a continuous variable in the multivariate analysis. The association between low PNI and an increased risk of postoperative AKI was better explained in the text: “Low preoperative PNI was significantly associated with an increased risk of postoperative AKI (OR: 0.97, 95% CI: 0.95–0.99, p=0.001).” (page 6, lines 166-167).

  1. Again, the regression for 1-year survival is confusing based on perceived definition of 1-year survival. Would also further describe these associations. PNI not just associated with survival, but how? Lower or higher PNI with lower or higher survival.

Response: The sentence was clarified as follows: “In the multivariate logistic regression analysis of 1-year mortality, low preoperative PNI (OR: 0.89, 95% CI: 0.84–0.95, p<0.001), DM (OR: 3.58, 95% CI: 1.50–8.56, p=0.004), and operation time (OR: 1.01, 95% CI: 1.00–1.01, p=0.044) were significantly associated with higher 1-year mortality (Table 3).” (page 7, lines 178-181).

  1. For the regression reported in Table 4, what other “potentially confounding variables” were controlled for?

Response: Potential confounders included age, sex, ASA, DM, CVA, tumor invasion, lymph node  invasion, distant metastasis, operation time, lowest MBP, total fluids administered, and RBC transfusion (legend of Table 4, page 9, lines 199-200).

  1. For the Kaplan Meier, how long were these patients followed for? What is the breakdown for overall follow up time per quartile?

Response: Our patients were followed for 1 year or until the end of the study to calculate 1-year mortality (Figure 2A) and overall mortality (Figure 2B).

  1. Is more granular data available regarding postoperative complications? I think that is likely more meaningful than ICU days or admission. Certainly specific complications of great interest to colorectal surgeons would include things such as surgical site infections, ileus, postoperative leak, return to operating room, 30-day readmission. These would also make sense to be associated with poor nutritional status.

Response: We agree with you. Postoperative complications included cardiac and pulmonary problems, bleeding, and leakage. This information was added to Table 1. However, we did not analyze the association between preoperative PNI and complications because the number of events was small.

  1. Is there information on stoma creation rates? Rates of rectal vs colon cancer? LAR vs APR?

Response: The rates of colon and rectal cancer were included in Table 1. There was no significant difference in these rates according to the PNI quartile. However, stoma creation rates were not calculated, which is a limitation of this study. For this reason, data on some surgical outcomes should be interpreted with caution. These limitations were discussed in the article: “Stoma creation rates were not included in the multivariable analysis. However, the impact of this confounding factor was reduced by adjusting for variables that could potentially affect the outcome, such as tumor staging and tumor location.(page 11, lines 289-291).

Discussion

  1. As these were cancer patients, data regarding clinical and/or pathologic stage seems important as this could be a major confounding factor and clearly higher stage patients are probably more likely to be more nutritionally deplete as well as more likely to die due to disease.

Response: Thank you for pointing out this issue. Data on cancer stage (tumor invasion, lymph node invasion, distant metastasis) were analyzed and included in Table 1 because these parameters are major potential confounders and could have a significant impact on the findings. These variables were included in the regression analysis of AKI and 1-year mortality (Tables 2 and 3). Tumor location, tumor invasion, lymph node invasion, and distant metastasis were not significantly associated with postoperative AKI and 1-year mortality.

Reviewer 2 Report

The authors have conducted a large retrospective cohort study on patients operated for colorectal cancer at a single centre 2008-2012. The primary outcome is acute kidney injury, while the exposure is a nutritional index. The main finding is that AKI was increased in patients with malnutrition, and that the latter was an independent predictor.

Introduction

  1. Short and concise. The aim of the study is clear – this is a prediction study without claims to causality.

Methods

  1. Why have the authors used such an early time period? Has the database not been updated?
  2. The authors have done a good job of defining the study population, including reasonable exclusion criteria
  3. It is a bit puzzling to me that an entire paragraph is devoted to anesthesia methods, while no mention is made about surgical approach. I suggest that the former could be shortened, while inserting some mention about surgery (especially stoma use, type of surgery/resection etc)
  4. This is a colorectal cancer study, but key variables are missing such as tumor stage, tumor location, type of surgical resection, defunctioning stoma use; in fact, some of these variables are so important (stoma use, type of resection and tumor location) that their omission invalidates the study results, at least partly
  5. The primary outcome definition is a bit unclear – I suppose that postoperative AKI is defined according to raised creatinine (as explained by the authors), but it does not say explicitly in which time period (though this should be 7 days, but this needs to be clarified at section 2.4).
  6. The statistical section needs improvement. As this is a prediction study, there are more modern methods available, e.g. LASSO, and reporting standards also, e.g. TRIPOD guidelines. There is also no mention of missing data and how this is handled.

Results

  1. My suggestion is to omit p values entirely in Table 1, per the STROBE guidelines. One reason is that this study is quite large, which will make even minute differences "significant" (though clinically irrelevant)
  2. Several important variables are missing, as already mentioned, in Table 1 and hopefully this could be revised/amended
  3. For me, there are some major issues here with the secondary outcomes/results about complications and survival. I find it utterly strange to see that complication rates are 1.2% in the Q1 group (even excluding AKI). In general, colorectal cancer operations render rates in the ballpark of 30-40%. The same holds true about survival rates (this is surely an error, must be mortality instead here – even if so, the rates seem unnaturally low, are these relative survival rates?)
  4. It seems that 1-year survival and overall survival are different estimates here, but it does not say the follow-up time for overall survival. Moreover, are the estimates calculated using formal survival analysis?

Discussion

1. The main issue with this study is that there are key confounders missing. We have no data on tumor stage, type of surgery, stoma use, etc.. this invalidates the findings and for this study to be published, such data is likely necessary. This should at least be specifically mentioned as limitations.

Author Response

Responses to Reviewer #2

Introduction

  1. Short and concise. The aim of the study is clear – this is a prediction study without claims to causality.

Response: Thank you for the remark. This study analyzed the association between preoperative PNI and postoperative AKI in colorectal cancer surgery.

Methods

  1. Why have the authors used such an early time period? Has the database not been updated?

Response: We appreciate the question. The database has not been updated.

  1. The authors have done a good job of defining the study population, including reasonable exclusion criteria

Response: Thank you for the compliment.

  1. It is a bit puzzling to me that an entire paragraph is devoted to anesthesia methods, while no mention is made about surgical approach. I suggest that the former could be shortened, while inserting some mention about surgery (especially stoma use, type of surgery/resection etc)

Response: We agree with you. We shortened the description of anesthetic techniques in Section 2.2 and added more information and references on surgical methods that might affect hemodynamics: “Regardless of the surgical approach, all patients underwent general anesthesia (open vs. laparoscopy). All surgeries were performed by a qualified and experienced surgical team. Conventional and laparoscopic surgeries were performed using standard protocols [19]. For laparoscopic surgery, pneumoperitoneum with carbon dioxide at 10–15 mmHg was established. Crystalloid and colloid solutions were administered during surgery. The total volume of infused synthetic colloids did not exceed 20 mL kg-1. During surgery, packed red blood cells (RBC) were transfused if plasma hemoglobin level was less than 8 g dL-1, and inotropes or vasopressors were administered if the mean blood pressure was less than 65 mmHg.” (page 2, lines 67-76).

  1. This is a colorectal cancer study, but key variables are missing such as tumor stage, tumor location, type of surgical resection, defunctioning stoma use; in fact, some of these variables are so important (stoma use, type of resection and tumor location) that their omission invalidates the study results, at least partly

Response: Data on tumor location, tumor invasion, lymph node invasion, and distant metastasis were analyzed and added to Table 1. In addition, these variables were included in the regression analysis of AKI and 1-year mortality (Tables 2 and 3). Tumor location, tumor invasion, lymph node invasion, and distant metastasis were not significantly associated with postoperative AKI and 1-year mortality. However, stoma creation rates were not calculated, which is a limitation of this study. For this reason, data on some surgical outcomes should be interpreted with caution. This limitation was discussed in the manuscript: “Stoma creation rates were not included in the multivariable analysis. However, the impact of this confounding factor was reduced by adjusting for variables that could potentially affect the outcome, such as tumor staging and tumor location.” (page 11, lines 289-291).

  1. The primary outcome definition is a bit unclear – I suppose that postoperative AKI is defined according to raised creatinine (as explained by the authors), but it does not say explicitly in which time period (though this should be 7 days, but this needs to be clarified at section 2.4)

Response: Postoperative AKI was defined according to KDIGO criteria, and this section was improved as follows: “The primary outcome was the incidence of postoperative AKI as defined by KDIGO criteria according to preoperative PNI quartiles.” (page 3, lines 104-105).

  1. The statistical section needs improvement. As this is a prediction study, there are more modern methods available, e.g. LASSO, and reporting standards also, e.g. TRIPOD guidelines. There is also no mention of missing data and how this is handled

Response: Thank you for raising this key issue. The purpose of this study was not to investigate the predictive power of PNI for AKI development in colorectal cancer surgery but to assess the association between PNI and AKI. A total of 193 patients had missing data and were excluded from the study. However, the patient exclusion did not significantly affect the results, as shown in the table below.

Final patients

Excluded patients

Total

p

(N=3,543)

(N=193)

(N=3736)

Age; year

59.8 ± 11.2

58.9 ± 8.5

59.7 ± 11.0

0.196

Sex; male

2176 (61.4%)

115 (59.6%)

2291 (61.3%)

0.665

Height                                             

1.6 ± 0.1

1.6 ± 0.1

1.6 ± 0.1

0.994

Weight                                                 

62.9 ± 10.5

60.5 ± 11.0

62.7 ± 10.5

0.002

BMI                                                

23.8 ± 3.1

23.0 ± 3.5

23.8 ± 3.1

0.001

DM

523 (14.8%)

21 (10.9%)

544 (14.6%)

0.166

HTN

1174 (33.1%)

58 (30.1%)

1232 (33.0%)

0.419

CVA

71 (2.0%)

1 (0.5%)

72 (1.9%)

0.182

ASA status

0.651

ASA 1

840 (23.7%)

40 (20.7%)

880 (23.6%)

ASA 2

2628 (74.2%)

149 (77.2%)

2777 (74.3%)

ASA 3

75 (2.1%)

4 (2.1%)

79 (2.1%)

Laparoscopic surgery

974 (27.5%)

49 (25.4%)

1023 (27.3%)

0.579

Colloid use

2551 (72.0%)

151 (78.2%)

2702 (72.3%)

0.071

Diuretics

18 (0.5%)

5 (2.6%)

23 (0.6%)

0.005

RBC transfusion                                          

0.1 ± 0.4

0.3 ± 1.2

0.1 ± 0.5

0.021

Urine output; mL kg-1 hr-1

1.9 ± 1.6

1.8 ± 1.6

1.9 ± 1.6

0.345

Lowest MBP; mmHg

71.0 ± 8.9

70.0 ± 9.4

71.0 ± 8.9

0.106

Operation time; min

171.1 ± 61.0

199.7 ± 93.5

172.5 ± 63.3

< 0.001

ICU admission

92 (2.6%)

6 (3.1%)

98 (2.6%)

0.840

Hospital days                                      

8.1 ± 6.5

10.5 ± 7.8

8.4 ± 7.3

< 0.001

Overall mortality

39 (1.1%)

4 (2.1%)

43 (1.2%)

0.281

1-year mortality

25 (0.7%)

3 (1.6%)

28 (0.7%)

0.174

This potential bias was discussed in the article: “A total of 193 patients with missing data were excluded from the study.” (page 11, line 288).

Results

  1. My suggestion is to omit p values entirely in Table 1, per the STROBE guidelines. One reason is that this study is quite large, which will make even minute differences "significant" (though clinically irrelevant)

Response: P-values were excluded from Table 1, and standardized mean difference values were added.

  1. Several important variables are missing, as already mentioned, in Table 1 and hopefully this could be revised/amended

Response: Thank you for the assessment. Data on tumor location, tumor invasion, lymph node invasion, and distant metastasis were analyzed and included in Table 1. Furthermore, these variables were included in the regression analysis of AKI and 1-year mortality (Tables 2 and 3). Tumor location, tumor invasion, lymph node invasion, and distant metastasis were not significantly associated with postoperative AKI and 1-year mortality.

     3. For me, there are some major issues here with the secondary outcomes/results about complications and survival. I find it utterly strange to see that complication rates are 1.2% in the Q1 group (even excluding AKI). In general, colorectal cancer operations render rates in the ballpark of 30-40%. The same holds true about survival rates (this is surely an error, must be mortality instead here – even if so, the rates seem unnaturally low, are these relative survival rates?)

Response: Thank you for the information. Asan Medical Center in Korea performed approximately 20,000 colorectal cancer surgeries until 2013, including 9100 cases of rectal cancer and 4600 and 6300 cases of right and left colon cancer, respectively. The 5-year survival rate of early and advanced rectal cancer was 94.1% and 80.6%, respectively. Therefore, if both rectal and colon cancer surgeries are considered, and if the follow-up period is shorter than 5 years, the survival rate is expected to be higher.

  1. Yang KM, Park IJ, Lee JL, et al. Does the different locations of colon cancer affect the oncologic outcome? A propensity-score matched analysis. Ann Coloproctol. 2019;35:15-23.
  2. https://eng.amc.seoul.kr/gb/lang/specialities/illness.do?code=CA_Colorectal_Cancer

  1. It seems that 1-year survival and overall survival are different estimates here, but it does not say the follow-up time for overall survival. Moreover, are the estimates calculated using formal survival analysis

Response: We changed the word “survival” to “mortality” throughout the manuscript. One-year mortality was calculated from the date of surgery to l-year follow-up, and overall mortality was determined from the date of surgery to the last follow-up. The text was revised as follows: “Data on the length of hospital stay, postoperative ICU admission, prolonged ICU stay (more than 2 days), postoperative complications, 1-year mortality (calculated from the date of surgery to 1-year follow-up), and overall mortality (determined from the date of surgery to the last follow-up) were also collected.” (page 3, lines 98-102).

Discussion

  1. The main issue with this study is that there are key confounders missing. We have no data on tumor stage, type of surgery, stoma use, etc.. this invalidates the findings and for this study to be published, such data is likely necessary. This should at least be specifically mentioned as limitations.

Response: Thank you for the assessment. Data on tumor location, tumor invasion, lymph node invasion, and distant metastasis were analyzed and included in Table 1. Moreover, these variables were included in the regression analysis of AKI and 1-year mortality (Tables 2 and 3). Tumor location, tumor invasion, lymph node invasion, and distant metastasis were not significantly associated with postoperative AKI and 1-year mortality. However, stoma creation rates were not calculated, which is a limitation of this study. For this reason, data on some surgical outcomes should be interpreted with caution. This limitation was discussed in the text: “Stoma creation rates were not included in the multivariable analysis. However, the impact of this confounding factor was reduced by adjusting for variables that could potentially affect the outcome, such as tumor staging and tumor location.” (page 11, lines 289-291).

Reviewer 3 Report

I have reviewed the manuscript “Association of preoperative prognostic nutritional index and 2 postoperative acute kidney injury in patients with colorectal 3 cancer surgery”. The data is valuable, the study subject is clinically relevant, and the manuscript is overall well-written. Yet, I have methodological concerns regarding the analyses performed and other important concerns regarding interpretation of the results and its clinical applications thereof.

My major concern is that it is not clear to me whether the current study is etiological in nature or not, which needs to be separated from prediction research (10.1093/ndt/gfw459). Please realize that prediction research is a distinct field of epidemiologic research aimed at predicting the risk of an outcome according to a model of statistically significant predictors, which not necessarily represents causal associations, etiological studies aim to understand a certain pathway of a disease in an attempt to prevent its onset or progression. This is particularly relevant taking into account that patients in the lowest PNI quartile are overall less healthy than other patients (128-139). I am wondering whether the association between PNI and AKI is rather a reflection of overall health rather than nutrition, inflammation, and immune status, as proposed by the authors. I think the authors should additionally provide multivariable-adjusted analyses accounting for all the variables that showed statistical difference over increasing quartiles of PNI.

Is the model still independent after controlling for this potential confounders?

Please also realize that the actual conclusion of this manuscript are in terms of prediction, yet the authors did not actually performed prediction analyses (area under the curve ROC), nor the prediction ability of PNI was compared with other usually common predictors in this specific clinical setting; which leaves the reader with little to grasp from this study.

This issue should be appropriately revised before further consideration of this manuscript for publication.

Please further see my comments and suggestions to revise this manuscript:

-Please discuss about potential bias induced by excluding patients with incomplete data or laboratory values.

-I think the study population is really large so that ANOVA can be used instead of t-test. Similarly, I would suggest Chi-square instead of Fisher’s.

-For prospective analyses, I think authors should first provide rather exploratory, univariable analyses (i.e., Kaplan-Meier), and only after that it makes sense to perform uni and multivariable analyses (logistic regression). Kaplan-Meier analyses for all the binary outcomes, i.e. including the primary outcome AKI, should be provided.

-Did you used single or two-tailed p values < 0.05 to consider statistical significance?

-In figure 1, please report how many patients were excluded concerning each of the detailed exclusion criteria.

-Please revise the use of the word “survival” throughout the manuscript, I have the impression the authors actually mean the opposite “mortality” (lines 159-161: I see higher –not lower- figures for Q1; so I guess you mean the opposite to survival).

-The authors state that 1-year and overall survival records were collected, but I do not really understand from where is this data extracted. Is the survival data of patients automatically updated in the patient files of this healthcare center (even after discharge)?

Also, please report median (IRQ) of follow-up (time) and end date of follow-up to record “overall survival”.

Lines 100-106: I think the outcomes should be defined in terms of the events of interest, so I do not think it is correct to mix the definition of the outcomes with the description of the analytical methods (in this case, over quartiles) used to study those outcomes.

Lines 43-44: “there are only few studies regarding the association between preoperative PNI 43 and postoperative AKI development in colorectal cancer patients”. Please provide references to those studies.

Line 133: I think it is not necessary to report that there is a difference in PNI according to quartiles of PNI. This is expected.

Author Response

Responses to Reviewer #3

  1. My major concern is that it is not clear to me whether the current study is etiological in nature or not, which needs to be separated from prediction research (10.1093/ndt/gfw459). Please realize that prediction research is a distinct field of epidemiologic research aimed at predicting the risk of an outcome according to a model of statistically significant predictors, which not necessarily represents causal associations, etiological studies aim to understand a certain pathway of a disease in an attempt to prevent its onset or progression. This is particularly relevant taking into account that patients in the lowest PNI quartile are overall less healthy than other patients (128-139). I am wondering whether the association between PNI and AKI is rather a reflection of overall health rather than nutrition, inflammation, and immune status, as proposed by the authors. I think the authors should additionally provide multivariable-adjusted analyses accounting for all the variables that showed statistical difference over increasing quartiles of PNI.

Is the model still independent after controlling for this potential confounders?

Please also realize that the actual conclusion of this manuscript are in terms of prediction, yet the authors did not actually performed prediction analyses (area under the curve ROC), nor the prediction ability of PNI was compared with other usually common predictors in this specific clinical setting; which leaves the reader with little to grasp from this study

Response: Thank you for the insightful comments. The purpose of this study was not to investigate the predictive power of PNI for AKI development in colorectal cancer surgery but to assess the association between PNI and AKI. The impact of confounding factors was reduced in the multivariate analysis by adjusting for variables that could potentially affect the outcome, such as tumor staging and tumor location. Even after controlling for potential confounders, there was a significant association between preoperative PNI and AKI development.

  1. Please discuss about potential bias induced by excluding patients with incomplete data or laboratory values.

Response: A total of 193 patients had missing data and were excluded from the study. However, the patient exclusion did not significantly affect the results, as shown in the table below.

Final patients

Excluded patients

Total

p

(N=3,543)

(N=193)

(N=3736)

Age; year

59.8 ± 11.2

58.9 ± 8.5

59.7 ± 11.0

0.196

Sex; male

2176 (61.4%)

115 (59.6%)

2291 (61.3%)

0.665

Height                                             

1.6 ± 0.1

1.6 ± 0.1

1.6 ± 0.1

0.994

Weight                                                 

62.9 ± 10.5

60.5 ± 11.0

62.7 ± 10.5

0.002

BMI                                                

23.8 ± 3.1

23.0 ± 3.5

23.8 ± 3.1

0.001

DM

523 (14.8%)

21 (10.9%)

544 (14.6%)

0.166

HTN

1174 (33.1%)

58 (30.1%)

1232 (33.0%)

0.419

CVA

71 (2.0%)

1 (0.5%)

72 (1.9%)

0.182

ASA status

0.651

ASA 1

840 (23.7%)

40 (20.7%)

880 (23.6%)

ASA 2

2628 (74.2%)

149 (77.2%)

2777 (74.3%)

ASA 3

75 (2.1%)

4 (2.1%)

79 (2.1%)

Laparoscopic surgery

974 (27.5%)

49 (25.4%)

1023 (27.3%)

0.579

Colloid use

2551 (72.0%)

151 (78.2%)

2702 (72.3%)

0.071

Diuretics

18 (0.5%)

5 (2.6%)

23 (0.6%)

0.005

RBC transfusion                                          

0.1 ± 0.4

0.3 ± 1.2

0.1 ± 0.5

0.021

Urine output; mL kg-1 hr-1

1.9 ± 1.6

1.8 ± 1.6

1.9 ± 1.6

0.345

Lowest MBP; mmHg

71.0 ± 8.9

70.0 ± 9.4

71.0 ± 8.9

0.106

Operation time; min

171.1 ± 61.0

199.7 ± 93.5

172.5 ± 63.3

< 0.001

ICU admission

92 (2.6%)

6 (3.1%)

98 (2.6%)

0.840

Hospital days                                      

8.1 ± 6.5

10.5 ± 7.8

8.4 ± 7.3

< 0.001

Overall mortality

39 (1.1%)

4 (2.1%)

43 (1.2%)

0.281

1-year mortality

25 (0.7%)

3 (1.6%)

28 (0.7%)

0.174

This potential bias was discussed in the text: “A total of 193 patients had missing data and were excluded from the study.(page 11, line 288).

  1. I think the study population is really large so that ANOVA can be used instead of t-test. Similarly, I would suggest Chi-square instead of Fisher’s.

Response: Thank you for the recommendations. ANOVA and the chi-square test were used in the study but were incorrectly described in the methods section. This section was revised as follows: “Categorical data were analyzed using the chi-square test, and continuous data were evaluated by analysis of variance (ANOVA).” (page 3, lines 110-111).

  1. For prospective analyses, I think authors should first provide rather exploratory, univariable analyses (i.e., Kaplan-Meier), and only after that it makes sense to perform uni and multivariable analyses (logistic regression). Kaplan-Meier analyses for all the binary outcomes, i.e. including the primary outcome AKI, should be provided.

Response: One-year survival and overall survival were calculated using the Kaplan-Meier method (Figure 2). Kaplan-Meier survival analysis was not performed on AKI and ICU admission because AKI and ICU admissions occurred immediately after surgery.

  1. Did you used single or two-tailed p values < 0.05 to consider statistical significance?

Response: Two-tailed p-values were used.

  1. In figure 1, please report how many patients were excluded concerning each of the detailed exclusion criteria.

Response: This information is shown in Figure 1.

  1. Please revise the use of the word “survival” throughout the manuscript, I have the impression the authors actually mean the opposite “mortality” (lines 159-161: I see higher –not lower- figures for Q1; so I guess you mean the opposite to survival).

Response: The word “survival” was replaced with “mortality” throughout the manuscript.

  1. The authors state that 1-year and overall survival records were collected, but I do not really understand from where is this data extracted. Is the survival data of patients automatically updated in the patient files of this healthcare center (even after discharge)?

Response: In Korea, all citizens are required to sign up for a national health insurance system, and when a patient dies, the death report is included in the database. In addition, patient survival data are automatically updated in the patient medical records in our center, accompanied by loss of health insurance eligibility.

  1. Also, please report median (IRQ) of follow-up (time) and end date of follow-up to record “overall survival”.

Response: Thank you for the suggestion. These data were incorporated in the text: “Mortality was followed for 3.14 (2.24-3.93) years, and the last follow-up was March 26, 2013.” (page 6, lines 164-165).

  1. Lines 100-106: I think the outcomes should be defined in terms of the events of interest, so I do not think it is correct to mix the definition of the outcomes with the description of the analytical methods (in this case, over quartiles) used to study those outcomes.

Response: As per your recommendation, the text was improved as follows: “The primary outcome was the incidence of postoperative AKI as defined by KDIGO criteria according to preoperative PNI levels. The secondary outcomes were the length of hospital stay, postoperative ICU admission, prolonged ICU stay (more than 2 days), postoperative complications (cardiac and pulmonary problems, bleeding, and leakage), 1-year mortality, and overall mortality according to preoperative PNI quartiles.” (page 3, lines 104-108).

  1. Lines 43-44: “there are only few studies regarding the association between preoperative PNI 43 and postoperative AKI development in colorectal cancer patients”. Please provide references to those studies.

Response: To our knowledge, no studies have evaluated the association between preoperative PNI and postoperative AKI development in patients with colorectal cancer. This section of the manuscript was revised as follows: “However, to our knowledge, no studies have evaluated the association between preoperative PNI and postoperative AKI development in colorectal cancer patients.” (page 2, lines 48-49).

  1. Line 133: I think it is not necessary to report that there is a difference in PNI according to quartiles of PNI. This is expected.

Response: We appreciate the suggestion. The text was improved as follows: “The groups with lower preoperative PNI had significantly lower levels of hemoglobin, albumin, and creatinine, and significantly higher levels of NLR, PLR, and RDW (all at p<0.001) (Table 1).” (page 4, lines 135-137)

Round 2

Reviewer 1 Report

The authors did a nice job responding to all prior comments and queries.

Author Response

We sincerely appreciate your comments.

Reviewer 2 Report

Thank you for your responses to my comments. After seeing these and those of the other reviewers', a few key issues remain.

  1. Study aim and design. I was also under the impression that this was a prediction study, not a causal/etiological study, but the authors have commented that this indeed not a prediction study, but they also state that this is not a causal study either. This leaves us with a descriptive study (see Hernan Miguel, Data Science), but the wording of confounding still implies that a causal approach is taken. However, there are several methodological issues here (and confusion in the terminology) that do not conform to a causal study. The statistical approach by using only variables with low p values as candidates for a regression model has been proven to be an inappropriate way of variable selection in any type of study, while a causal type of study would use e.g. causal diagrams or at least prior knowledge of important confounders, notwithstanding statistical relationships within this particular data set.
  2. Missing data. By showing a table that there are few differences for those with missing and for those with complete data set, not much has been said. The pattern of missingness is the important part, and there might be systematic differences at play. There are standard imputation methods for use here.
  3. Complication rates. While survival rates seem reliable enough in comparison to other Korean work (impressive!), it seems all but impossible that in a cohort of more than 3000 patients you'd find only four cases of anastomotic leakage. This is completely unheard of. Are you really sure this is correct (the complication rates)? If this is indeed true, this is a subject for an entire new manuscript with potential major impact (I am not sarcastic), to understand how your centre delivers such quality of care.

Author Response

Responses to Reviewer #2

  1. Study aim and design. I was also under the impression that this was a prediction study, not a causal/etiological study, but the authors have commented that this indeed not a prediction study, but they also state that this is not a causal study either. This leaves us with a descriptive study (see Hernan Miguel, Data Science), but the wording of confounding still implies that a causal approach is taken. However, there are several methodological issues here (and confusion in the terminology) that do not conform to a causal study. The statistical approach by using only variables with low p values as candidates for a regression model has been proven to be an inappropriate way of variable selection in any type of study, while a causal type of study would use e.g. causal diagrams or at least prior knowledge of important confounders, notwithstanding statistical relationships within this particular data set.

Response: Thank you for raising this key issue. This study was not a causal/ etiological study. The purpose of this study was not to investigate the predictive power of PNI for AKI development in colorectal cancer surgery but to assess the association between PNI and AKI. We agree with your comments on the methodology. As per your recommendation and reviewer No. 3' comments, we included all the variables that showed statistical difference over increasing quartiles of PNI (SMD>0.1), variables with p<0.1 in univariate, and prior knowledge of important variables for AKI and mortality to the multivariable-adjusted analyses. The results are shown in the Table 2 and Table 3 below, and although all these variables were included, it did not significantly affect our previous results. We revised the previous sentences in result section as follows: “In the multivariate analysis, low preoperative PNI was significantly associated with an increased risk of postoperative AKI (OR: 0.96, 95% CI: 0.93–0.99, p=0.003). Additionally, male gender (OR 1.60, 95%CI 1.19–2.14, p<0.001), BMI (OR 1.06, 95%CI 1.02–1.11, p =0.003), DM (OR 1.48, 95%CI 1.10–1.98, p=0.009), HTN (OR 1.46, 95%CI 1.12–1.91, p=0.006), white blood cell (OR 1.10, 95%CI 1.02–1.19, p=0.016), and creatinine (OR 0.05, 95%CI 0.02–0.13, p<0.001) were significantly associated with postoperative AKI (Table 2).” (page 6, lines 163-168), “In the multivariate logistic regression analysis of 1-year mortality, low preoperative PNI (OR: 0.92, 95% CI: 0.86–0.98, p=0.011), DM (OR: 3.91, 95% CI: 1.60–9.60, p=0.003), and smoking (OR: 4.12, 95% CI: 1.28–13.31, p=0.018) were significantly associated with higher 1-year mortality (Table 3).” (page 7, lines 176-179).

Table 2. Univariate and multivariate logistic regression analysis of AKI.

Univariate

Multivariate

OR

95% CI

P-value

OR

95% CI

P-value

PNI

0.97

0.96–0.99

0.003

0.96

0.93–0.99

0.003

Age

1.01

1.00–1.02

0.018

1.00

0.99–1.02

0.465

Sex (male)

1.65

1.30–1.21

<.001

1.60

1.19–2.14

<.001

BMI

1.04

1.00–1.07

0.042

1.06

1.02–1.11

0.003

DM

1.83

1.40–2.38

<.001

1.48

1.10–1.98

0.009

HTN

1.50

1.20–1.87

<.001

1.46

1.12–1.91

0.006

CVA

1.29

0.63–2.61

0.485

Smoking

1.43

1.13–1.80

0.003

1.09

0.81–1.45

0.573

ASA

0.010

0.130

ASA status 1

1.00 (Ref.)

1.00 (Ref.)

ASA status 2

1.20

0.91–1.56

0.195

0.80

0.58–1.11

0.189

ASA status 3

2.59

1.40–4.78

0.002

1.36

0.68–2.73

0.388

Tumor location

colon

1.00 (Ref.)

1.00 (Ref.)

rectum

0.90

0.70–1.14

0.376

Tumor invasion

Tis, T1, T2

1.00 (Ref.)

1.00 (Ref.)

T3, T4

1.21

0.93–1.58

0.153

1.12

0.81–1.54

0.497

Lymph node invasion

N0

1.00 (Ref.)

1.00 (Ref.)

N1, N2

1.17

0.94–1.46

0.152

1.02

0.79–1.33

0.862

Distant metastasis

M0

1.00 (Ref.)

1.00 (Ref.)

M1

1.28

0.91–1.79

0.157

1.09

0.74–1.58

0.672

Operation time; min

1.00

1.00–1.00

0.333

1.00

1.00–1.00

0.798

Laparoscopic surgery

0.98

0.77–1.25

0.878

1.05

0.80–1.39

0.712

Lowest MBP; mmHg

1.00

0.99–1.02

0.625

1.00

0.99–1.02

0.496

Total fluids; mL kg-1

1.00

0.99–1.01

0.565

1.00

0.99–1.02

0.748

Crystalloid; mL kg-1

1.00

1.00–1.01

0.289

Colloid; mL kg-1

0.99

0.96–1.01

0.244

Synthetic Colloid use

0.91

0.72–1.16

0.464

0.98

0.75–1.27

0.862

Urine output; mL kg-1 hr-1

1.03

0.97–1.10

0.372

1.06

0.99–1.13

0.124

Diuretics

1.77

0.51–6.14

0.369

1.57

0.35-5.19

0.497

RBC transfusion

1.35

0.74–2.44

0.327

1.04

0.51–2.11

0.922

White blood cell

1.04

0.98–1.09

0.179

1.10

1.02–1.19

0.016

Hemoglobin

0.98

0.93–1.04

0.560

0.96

0.89–1.04

0.316

Albumin; g dL-1

0.55

0.44–0.69

<.001

Creatinine

0.38

0.21–0.71

0.002

0.05

0.02–0.13

<.001

eGFR

1.00

0.99–1.01

0.608

NLR

0.98

0.93–1.05

0.594

0.90

0.78–1.03

0.118

PLR

1.00

0.99–1.01

0.302

0.98

0.94–1.02

0.337

RDW

1.01

0.97–1.05

0.755

0.99

0.94–1.03

0.579

Table 3. Univariate and multivariate logistic regression analysis of 1-year mortality.

Univariate

Multivariate

OR

95% CI

P-value

OR

95% CI

P-value

PNI

0.87

0.82–0.92

<.001

0.92

0.86–0.98

0.011

Age

1.05

1.01–1.10

0.008

1.04

1.00–1.09

0.054

Sex (male)

1.12

0.49–2.54

0.790

1.50

0.41–5.50

0.545

BMI

0.90

0.79–1.02

0.111

1.02

0.89–1.18

0.760

DM

3.91

1.75–8.74

<.001

3.91

1.60–9.60

0.003

HTN

1.87

0.85–4.11

0.119

1.33

0.49–3.62

0.577

CVA

2.05

0.27–15.38

0.484

Smoking

2.96

1.3–6.72

0.009

4.12

1.28–13.31

0.018

ASA

<.001

0.130

ASA status 1

1.00 (Ref.)

1.00 (Ref.)

ASA status 2

1.82

0.53–6.21

0.342

0.56

0.14–2.28

0.418

ASA status 3

19.93

4.67–85.13

<.001

2.93

0.52–16.41

0.221

Tumor location

colon

1.00 (Ref.)

1.00 (Ref.)

rectum

1.39

0.62–3.13

0.430

Tumor invasion

Tis, T1, T2

1.00 (Ref.)

1.00 (Ref.)

T3, T4

7.49

1.02–54.90

0.049

3.58

0.38–33.67

0.265

Lymph node invasion

N0

1.00 (Ref.)

1.00 (Ref.)

N1, N2

2.89

1.15–7.24

0.025

1.81

0.69–4.78

0.230

Distant metastasis

M0

1.00 (Ref.)

1.00 (Ref.)

M1

2.94

1.17–7.38

0.022

1.67

0.56–4.99

0.358

Operation time; min

1.00

1.00–1.01

0.022

1.01

1.00–1.01

0.116

Laparoscopic surgery

0.11

0.01–0.81

0.030

0.15

0.02–1.16

0.070

Lowest MBP; mmHg

0.95

0.90–0.99

0.027

1.00

0.94–1.05

0.853

Total fluids; mL kg-1

1.02

1.00–1.04

0.030

1.27

0.90–1.80

0.178

Crystalloid; mL kg-1

1.03

1.01–1.05

0.005

0.80

0.56–1.13

0.208

Colloid; mL kg-1

0.94

0.86–1.03

0.167

0.72

0.49–1.04

0.080

Synthetic Colloid use

0.42

0.19–0.92

0.030

0.40

0.16–1.02

0.054

Urine output; mL kg-1 hr-1

0.85

0.63–1.16

0.305

Diuretics

*

0.992

RBC transfusion

3.07

0.71–13.19

0.132

0.59

0.09–3.93

0.593

White blood cell

1.13

0.96–1.34

0.142

1.06

0.87–1.30

0.547

Hemoglobin

0.7

0.58–0.85

<.001

0.92

0.70–1.20

0.534

Albumin; g dL-1

0.19

0.10–0.36

<.001

Creatinine

0.11

0.01–1.05

0.05503

0.15

0.01–3.27

0.228

eGFR

1.00

0.97–1.03

0.930

NLR

1.10

1.02–1.18

0.010

0.98

0.73–1.31

0.888

PLR

1.02

1.00–1.05

0.003

1.00

0.92–1.09

0.952

RDW

1.19

1.09–1.31

<.001

1.08

0.95–1.22

0.256

  1. Missing data. By showing a table that there are few differences for those with missing and for those with complete data set, not much has been said. The pattern of missingness is the important part, and there might be systematic differences at play. There are standard imputation methods for use here.

Response: Thank you for the assessment. We are cautiously speculating that you probably think that patients with missing data were included in analysis. However, in this study, patients with any missing data were excluded, therefore, a total of 193 patients were excluded, and 3,543 patients were patients with complete data. Thus, no additional standard imputation methods were performed in our study. Excluded patients did not differ significantly in most variables from patients included in the study, as shown in the table below.

Supplementary table 1. Characteristics of patients excluded from the study

Final patients

Excluded patients

Total

p

(N=3,543)

(N=193)

(N=3736)

Age; year

59.8 ± 11.2

58.9 ± 8.5

59.7 ± 11.0

0.196

Sex; male

2176 (61.4%)

115 (59.6%)

2291 (61.3%)

0.665

Height                                             

1.6 ± 0.1

1.6 ± 0.1

1.6 ± 0.1

0.994

Weight                                                 

62.9 ± 10.5

60.5 ± 11.0

62.7 ± 10.5

0.002

BMI                                                

23.8 ± 3.1

23.0 ± 3.5

23.8 ± 3.1

0.001

DM

523 (14.8%)

21 (10.9%)

544 (14.6%)

0.166

HTN

1174 (33.1%)

58 (30.1%)

1232 (33.0%)

0.419

CVA

71 (2.0%)

1 (0.5%)

72 (1.9%)

0.182

ASA status

0.651

ASA 1

840 (23.7%)

40 (20.7%)

880 (23.6%)

ASA 2

2628 (74.2%)

149 (77.2%)

2777 (74.3%)

ASA 3

75 (2.1%)

4 (2.1%)

79 (2.1%)

Laparoscopic surgery

974 (27.5%)

49 (25.4%)

1023 (27.3%)

0.579

Colloid use

2551 (72.0%)

151 (78.2%)

2702 (72.3%)

0.071

Diuretics

18 (0.5%)

5 (2.6%)

23 (0.6%)

0.005

RBC transfusion                                          

0.1 ± 0.4

0.3 ± 1.2

0.1 ± 0.5

0.021

Urine output; mL kg-1 hr-1

1.9 ± 1.6

1.8 ± 1.6

1.9 ± 1.6

0.345

Lowest MBP; mmHg

71.0 ± 8.9

70.0 ± 9.4

71.0 ± 8.9

0.106

Operation time; min

171.1 ± 61.0

199.7 ± 93.5

172.5 ± 63.3

< 0.001

ICU admission

92 (2.6%)

6 (3.1%)

98 (2.6%)

0.840

Hospital days                                      

8.1 ± 6.5

10.5 ± 7.8

8.4 ± 7.3

< 0.001

Overall mortality

39 (1.1%)

4 (2.1%)

43 (1.2%)

0.281

1-year mortality

25 (0.7%)

3 (1.6%)

28 (0.7%)

0.174

This table was added to the supplementary material of the manuscript (supple. Table 1).

Although 193 patients were excluded due to the missing data, we cautiously think they would not have had a significant impact on our findings, because they accounted for approximately 5% of the total number.

  1. Madley-Dowd, P.; Hughes, R.; Tilling, K.; Heron, J. The proportion of missing data should not be used to guide decisions on multiple imputation. Journal of Clinical Epidemiology 2019, 110, 63-73, doi:https://doi.org/10.1016/j.jclinepi.2019.02.016.

Finally, we have added the comment on this issue to the Discussion section and described it as follows: “A total of 193 patients (5.2%) with missing data were excluded from the study, which may have affected surgical outcome. However, their characteristics were not significantly different from the patients who included in the study (supplementary table 1) and because of their small number, we don't think it would have had a significant impact on our findings.” (page 11, lines 286-290).

  1. Complication rates. While survival rates seem reliable enough in comparison to other Korean work (impressive!), it seems all but impossible that in a cohort of more than 3000 patients you'd find only four cases of anastomotic leakage. This is completely unheard of. Are you really sure this is correct (the complication rates)? If this is indeed true, this is a subject for an entire new manuscript with potential major impact (I am not sarcastic), to understand how your centre delivers such quality of care.Journal of Personalized MedicineThe occurrence of surgical site leakage previously submitted was described for those who had a record immediately after surgery or who had undergone reoperation through a chart review of our hospital's EMR. However, we have found that there is a significant error in this record. Patients with less severe symptoms were treated by using antibiotics without reoperation, or most leakage records are missing. Therefore, we decided to eliminate surgical site leakage from the results and add this to the limitation.

Response: Thank you for asking and we apologize for our errors in the record. The incidence of previously submitted surgical site leakage was described for those who had a record immediately after surgery or who had undergone reoperation through a chart review of our hospital's EMR. However, patients with less severe symptoms were treated by using antibiotics without reoperation, or most leakage records were missing. Therefore, we decided to remove surgical site leakage from the results and add this to the limitation section as follows: “In our study, stoma creation rates and surgical site leakage were also not analyzed. Therefore, careful interpretation of some surgical results is required.” (page 11, lines 290-291).

Reviewer 3 Report

Unfortunately, most of my major concerns seem to have been overlooked in this revision, as the analyses I requested were not performed. Likewise, the discussions I requested about issues important for the reader to consider in order to interpret the study correctly were not added to the manuscript.

  • My most important comment for the authors concerning the need to address the issue of clearly differentiating between an etiological and predictive study was not addressed at all. Nothing was changed in the manuscript to help the reader understand what kind of research was actually performed. The authors attempted to clarify this to me by pointing out that -to them- this is not a prediction study, but they should have taken into account that this issue is certainly not clear to me but also to reviewer #2, who stated that "this is a prediction study" clear in opposition to what the authors think it is clear. This should have been appropriately revised in the manuscript (not just in comment to Reviewers). Furthermore, because I was anticipating that the authors would consider this study as etiological by nature, I asked for adding all the variables that showed statistical difference over increasing quartiles of PNI to the multivariable-adjusted analyses, which was completely ignored by the authors. This leaves me at the same stage of the first round of revision, i.e., I wonder whether the reported association truly is causal and independent of confounding or not. I clearly asked “Is the model still independent after controlling for this potential confounders?”. Unfortunately, I could not evaluate this issue as the analyses are still not provided by the authors and the question remained unanswered, which makes me wonder why the authors avoided performing appropriate multivariable analyses.
  • Regarding my concern about missing data, the authors excluded patients with missing data and compared baseline characteristics between included and excluded patients. While this is an initial step towards evaluating potential bias induced by missing data –and, therefore, it should have been actually added to the supplemental material of the manuscript so that other readers with the same doubt as me could evaluate this issue-, it should be realized that this is truly not an established method for addressing the issue of missing data. Appropriate sensitivity analyses are still pending, and, as suggested in the first round of revision, discussion of this issue should still be added to the Discussion section of the manuscript.
  • Regarding response to comment 4, I’m afraid this is not a justification for not performing Kaplan-Meier analyses of the primary outcome of the study, for instance, with the x-axis on a days scale.
  • Regarding comment #10. I do not think the authors understood my point. The revised text is still describing the outcomes in terms of PNI quartiles, which is not correct, as this is the method used to analyze the outcomes, not the outcomes by definition.
  • Also important, some other minor comments that the authors did attend by means of replying in text to me, unfortunately, were not accompanied by the corresponding revision of the manuscript, which is a partial revision (comments #5 and 8), as this may frustrate readers with the same potential questions and concerns I raised in the first round of revision.

Author Response

Responses to Reviewer #3

  1. My most important comment for the authors concerning the need to address the issue of clearly differentiating between an etiological and predictive study was not addressed at all. Nothing was changed in the manuscript to help the reader understand what kind of research was actually performed. The authors attempted to clarify this to me by pointing out that -to them- this is not a prediction study, but they should have taken into account that this issue is certainly not clear to me but also to reviewer #2, who stated that "this is a prediction study" clear in opposition to what the authors think it is clear. This should have been appropriately revised in the manuscript (not just in comment to Reviewers). Furthermore, because I was anticipating that the authors would consider this study as etiological by nature, I asked for adding all the variables that showed statistical difference over increasing quartiles of PNI to the multivariable-adjusted analyses, which was completely ignored by the authors. This leaves me at the same stage of the first round of revision, i.e., I wonder whether the reported association truly is causal and independent of confounding or not. I clearly asked “Is the model still independent after controlling for this potential confounders?”. Unfortunately, I could not evaluate this issue as the analyses are still not provided by the authors and the question remained unanswered, which makes me wonder why the authors avoided performing appropriate multivariable analyses.

Response: Thank you for raising this key issue. This study was not a causal/ etiological study. The purpose of this study was not to investigate the predictive power of PNI for AKI development in colorectal cancer surgery but to assess the association between PNI and AKI. We agree with your opinion that there is an error in the selection of variables in the logistic analysis method. As per your recommendation and reviewer No. 2' comments, we included all the variables that showed statistical difference over increasing quartiles of PNI (SMD>0.1), variables with p<0.1 in univariate, and prior knowledge of important variables for AKI and mortality to the multivariable-adjusted analyses. The results are shown in the Table 2 and Table 3 below, and although all these variables were included, it did not significantly affect our previous results. We revised the previous sentences in result section as follows: “In the multivariate analysis, low preoperative PNI was significantly associated with an increased risk of postoperative AKI (OR: 0.96, 95% CI: 0.93–0.99, p=0.003). Additionally, male gender (OR 1.60, 95%CI 1.19–2.14, p<0.001), BMI (OR 1.06, 95%CI 1.02–1.11, p =0.003), DM (OR 1.48, 95%CI 1.10–1.98, p=0.009), HTN (OR 1.46, 95%CI 1.12–1.91, p=0.006), white blood cell (OR 1.10, 95%CI 1.02–1.19, p=0.016), and creatinine (OR 0.05, 95%CI 0.02–0.13, p<0.001) were significantly associated with postoperative AKI (Table 2).” (page 6, lines 163-168), “In the multivariate logistic regression analysis of 1-year mortality, low preoperative PNI (OR: 0.92, 95% CI: 0.86–0.98, p=0.011), DM (OR: 3.91, 95% CI: 1.60–9.60, p=0.003), and smoking (OR: 4.12, 95% CI: 1.28–13.31, p=0.018) were significantly associated with higher 1-year mortality (Table 3).” (page 7, lines 176-179).

Table 2. Univariate and multivariate logistic regression analysis of AKI.

Univariate

Multivariate

OR

95% CI

P-value

OR

95% CI

P-value

PNI

0.97

0.96–0.99

0.003

0.96

0.93–0.99

0.003

Age

1.01

1.00–1.02

0.018

1.00

0.99–1.02

0.465

Sex (male)

1.65

1.30–1.21

<.001

1.60

1.19–2.14

<.001

BMI

1.04

1.00–1.07

0.042

1.06

1.02–1.11

0.003

DM

1.83

1.40–2.38

<.001

1.48

1.10–1.98

0.009

HTN

1.50

1.20–1.87

<.001

1.46

1.12–1.91

0.006

CVA

1.29

0.63–2.61

0.485

Smoking

1.43

1.13–1.80

0.003

1.09

0.81–1.45

0.573

ASA

0.010

0.130

ASA status 1

1.00 (Ref.)

1.00 (Ref.)

ASA status 2

1.20

0.91–1.56

0.195

0.80

0.58–1.11

0.189

ASA status 3

2.59

1.40–4.78

0.002

1.36

0.68–2.73

0.388

Tumor location

colon

1.00 (Ref.)

1.00 (Ref.)

rectum

0.90

0.70–1.14

0.376

Tumor invasion

Tis, T1, T2

1.00 (Ref.)

1.00 (Ref.)

T3, T4

1.21

0.93–1.58

0.153

1.12

0.81–1.54

0.497

Lymph node invasion

N0

1.00 (Ref.)

1.00 (Ref.)

N1, N2

1.17

0.94–1.46

0.152

1.02

0.79–1.33

0.862

Distant metastasis

M0

1.00 (Ref.)

1.00 (Ref.)

M1

1.28

0.91–1.79

0.157

1.09

0.74–1.58

0.672

Operation time; min

1.00

1.00–1.00

0.333

1.00

1.00–1.00

0.798

Laparoscopic surgery

0.98

0.77–1.25

0.878

1.05

0.80–1.39

0.712

Lowest MBP; mmHg

1.00

0.99–1.02

0.625

1.00

0.99–1.02

0.496

Total fluids; mL kg-1

1.00

0.99–1.01

0.565

1.00

0.99–1.02

0.748

Crystalloid; mL kg-1

1.00

1.00–1.01

0.289

Colloid; mL kg-1

0.99

0.96–1.01

0.244

Synthetic Colloid use

0.91

0.72–1.16

0.464

0.98

0.75–1.27

0.862

Urine output; mL kg-1 hr-1

1.03

0.97–1.10

0.372

1.06

0.99–1.13

0.124

Diuretics

1.77

0.51–6.14

0.369

1.57

0.35-5.19

0.497

RBC transfusion

1.35

0.74–2.44

0.327

1.04

0.51–2.11

0.922

White blood cell

1.04

0.98–1.09

0.179

1.10

1.02–1.19

0.016

Hemoglobin

0.98

0.93–1.04

0.560

0.96

0.89–1.04

0.316

Albumin; g dL-1

0.55

0.44–0.69

<.001

Creatinine

0.38

0.21–0.71

0.002

0.05

0.02–0.13

<.001

eGFR

1.00

0.99–1.01

0.608

NLR

0.98

0.93–1.05

0.594

0.90

0.78–1.03

0.118

PLR

1.00

0.99–1.01

0.302

0.98

0.94–1.02

0.337

RDW

1.01

0.97–1.05

0.755

0.99

0.94–1.03

0.579

Table 3. Univariate and multivariate logistic regression analysis of 1-year mortality.

Univariate

Multivariate

OR

95% CI

P-value

OR

95% CI

P-value

PNI

0.87

0.82–0.92

<.001

0.92

0.86–0.98

0.011

Age

1.05

1.01–1.10

0.008

1.04

1.00–1.09

0.054

Sex (male)

1.12

0.49–2.54

0.790

1.50

0.41–5.50

0.545

BMI

0.90

0.79–1.02

0.111

1.02

0.89–1.18

0.760

DM

3.91

1.75–8.74

<.001

3.91

1.60–9.60

0.003

HTN

1.87

0.85–4.11

0.119

1.33

0.49–3.62

0.577

CVA

2.05

0.27–15.38

0.484

Smoking

2.96

1.3–6.72

0.009

4.12

1.28–13.31

0.018

ASA

<.001

0.130

ASA status 1

1.00 (Ref.)

1.00 (Ref.)

ASA status 2

1.82

0.53–6.21

0.342

0.56

0.14–2.28

0.418

ASA status 3

19.93

4.67–85.13

<.001

2.93

0.52–16.41

0.221

Tumor location

colon

1.00 (Ref.)

1.00 (Ref.)

rectum

1.39

0.62–3.13

0.430

Tumor invasion

Tis, T1, T2

1.00 (Ref.)

1.00 (Ref.)

T3, T4

7.49

1.02–54.90

0.049

3.58

0.38–33.67

0.265

Lymph node invasion

N0

1.00 (Ref.)

1.00 (Ref.)

N1, N2

2.89

1.15–7.24

0.025

1.81

0.69–4.78

0.230

Distant metastasis

M0

1.00 (Ref.)

1.00 (Ref.)

M1

2.94

1.17–7.38

0.022

1.67

0.56–4.99

0.358

Operation time; min

1.00

1.00–1.01

0.022

1.01

1.00–1.01

0.116

Laparoscopic surgery

0.11

0.01–0.81

0.030

0.15

0.02–1.16

0.070

Lowest MBP; mmHg

0.95

0.90–0.99

0.027

1.00

0.94–1.05

0.853

Total fluids; mL kg-1

1.02

1.00–1.04

0.030

1.27

0.90–1.80

0.178

Crystalloid; mL kg-1

1.03

1.01–1.05

0.005

0.80

0.56–1.13

0.208

Colloid; mL kg-1

0.94

0.86–1.03

0.167

0.72

0.49–1.04

0.080

Synthetic Colloid use

0.42

0.19–0.92

0.030

0.40

0.16–1.02

0.054

Urine output; mL kg-1 hr-1

0.85

0.63–1.16

0.305

Diuretics

*

0.992

RBC transfusion

3.07

0.71–13.19

0.132

0.59

0.09–3.93

0.593

White blood cell

1.13

0.96–1.34

0.142

1.06

0.87–1.30

0.547

Hemoglobin

0.7

0.58–0.85

<.001

0.92

0.70–1.20

0.534

Albumin; g dL-1

0.19

0.10–0.36

<.001

Creatinine

0.11

0.01–1.05

0.05503

0.15

0.01–3.27

0.228

eGFR

1.00

0.97–1.03

0.930

NLR

1.10

1.02–1.18

0.010

0.98

0.73–1.31

0.888

PLR

1.02

1.00–1.05

0.003

1.00

0.92–1.09

0.952

RDW

1.19

1.09–1.31

<.001

1.08

0.95–1.22

0.256

  1. Regarding my concern about missing data, the authors excluded patients with missing data and compared baseline characteristics between included and excluded patients. While this is an initial step towards evaluating potential bias induced by missing data –and, therefore, it should have been actually added to the supplemental material of the manuscript so that other readers with the same doubt as me could evaluate this issue-, it should be realized that this is truly not an established method for addressing the issue of missing data. Appropriate sensitivity analyses are still pending, and, as suggested in the first round of revision, discussion of this issue should still be added to the Discussion section of the manuscript.

Thanks for pointing out an important issue. In this study, patients with any missing data were excluded, therefore, a total of 193 patients were excluded, and 3,543 patients were patients with complete data. Excluded patients did not differ significantly in most variables from patients included in the study, as shown in the table below. As per your recommendation, this table was added to the supplementary material of the manuscript (supple. Table 1).

Supplementary table 1. Characteristics of patients excluded from the study

Final patients

Excluded patients

Total

p

(N=3,543)

(N=193)

(N=3736)

Age; year

59.8 ± 11.2

58.9 ± 8.5

59.7 ± 11.0

0.196

Sex; male

2176 (61.4%)

115 (59.6%)

2291 (61.3%)

0.665

Height                                             

1.6 ± 0.1

1.6 ± 0.1

1.6 ± 0.1

0.994

Weight                                                 

62.9 ± 10.5

60.5 ± 11.0

62.7 ± 10.5

0.002

BMI                                                

23.8 ± 3.1

23.0 ± 3.5

23.8 ± 3.1

0.001

DM

523 (14.8%)

21 (10.9%)

544 (14.6%)

0.166

HTN

1174 (33.1%)

58 (30.1%)

1232 (33.0%)

0.419

CVA

71 (2.0%)

1 (0.5%)

72 (1.9%)

0.182

ASA status

0.651

ASA 1

840 (23.7%)

40 (20.7%)

880 (23.6%)

ASA 2

2628 (74.2%)

149 (77.2%)

2777 (74.3%)

ASA 3

75 (2.1%)

4 (2.1%)

79 (2.1%)

Laparoscopic surgery

974 (27.5%)

49 (25.4%)

1023 (27.3%)

0.579

Colloid use

2551 (72.0%)

151 (78.2%)

2702 (72.3%)

0.071

Diuretics

18 (0.5%)

5 (2.6%)

23 (0.6%)

0.005

RBC transfusion                                          

0.1 ± 0.4

0.3 ± 1.2

0.1 ± 0.5

0.021

Urine output; mL kg-1 hr-1

1.9 ± 1.6

1.8 ± 1.6

1.9 ± 1.6

0.345

Lowest MBP; mmHg

71.0 ± 8.9

70.0 ± 9.4

71.0 ± 8.9

0.106

Operation time; min

171.1 ± 61.0

199.7 ± 93.5

172.5 ± 63.3

< 0.001

ICU admission

92 (2.6%)

6 (3.1%)

98 (2.6%)

0.840

Hospital days                                      

8.1 ± 6.5

10.5 ± 7.8

8.4 ± 7.3

< 0.001

Overall mortality

39 (1.1%)

4 (2.1%)

43 (1.2%)

0.281

1-year mortality

25 (0.7%)

3 (1.6%)

28 (0.7%)

0.174

Although 193 patients were excluded due to the missing data, we cautiously think they would not have had a significant impact on our findings, because they accounted for approximately 5% of the total number.

  1. Madley-Dowd, P.; Hughes, R.; Tilling, K.; Heron, J. The proportion of missing data should not be used to guide decisions on multiple imputation. Journal of Clinical Epidemiology 2019, 110, 63-73, doi:https://doi.org/10.1016/j.jclinepi.2019.02.016.

Finally, we have added the comment on this issue to the Discussion section and described it as follows: “A total of 193 patients (5.2%) with missing data were excluded from the study, which may have affected surgical outcome. However, their characteristics were not significantly different from the patients who included in the study (supplementary table 1) and because of their small number, we don't think it would have had a significant impact on our findings.” (page 11, lines 286-290).

  1. Regarding response to comment 4, I’m afraid this is not a justification for not performing Kaplan-Meier analyses of the primary outcome of the study, for instance, with the x-axis on a days scale.

Response: As per your recommendation, we performed Kaplan-Meier survival analysis on AKI as below and added to Supplementary figure 1.

Supplementary figure 1. Kaplan–Meier curve for acute kidney injury incidence (log-rank test; p < 0.001).

  1. Regarding comment #10. I do not think the authors understood my point. The revised text is still describing the outcomes in terms of PNI quartiles, which is not correct, as this is the method used to analyze the outcomes, not the outcomes by definition.

Response: Thank you for the recommendations. We understood what you are pointing out and revised the previous sentence as follows “Additionally, we assessed surgical outcomes such as hospital stay, ICU admission, and postoperative complications.” (page 1, lines 18-19), “The primary outcome was the incidence of postoperative AKI as defined by KDIGO criteria. The secondary outcomes were the length of hospital stay, postoperative ICU admission, prolonged ICU stay (>2 days), postoperative complications (cardiac and pulmonary problems, bleeding, and surgical site leakage), 1-year mortality, and overall mortality.” (page 3, lines 104-107).

  1. Also important, some other minor comments that the authors did attend by means of replying in text to me, unfortunately, were not accompanied by the corresponding revision of the manuscript, which is a partial revision (comments #5 and 8), as this may frustrate readers with the same potential questions and concerns I raised in the first round of revision.

Response: As per your recommendation, revision to comments #5 have been added to the statistical section as follows: “We used two‐tailed P values in tests of significance.” (page 3, lines 119).

Revision to comments #8 have been added to the Discussion section as follows “In Korea, all citizens are required to sign up for a national health insurance system, and when a patient dies, the death report is included in the database. In addition, patient survival data are automatically updated in the patient medical records in our center, accompanied by loss of health insurance eligibility.” (page 11, lines 276-280).
